# Features of membrane protein sequence direct post-translational insertion

Ilya A. Kalinin[1,2], Hadas Peled-Zehavi [1,2], Alon B. D. Barshap[1], Shai A. Tamari [1], Yarden Weiss [1], Reinat Nevo [1] & Nir Fluman [1] ✉

The proper folding of multispanning membrane proteins (MPs) hinges on the accurate insertion of their transmembrane helices (TMs) into the membrane. Predominantly, TMs are inserted during protein translation, via a conserved mechanism centered around the Sec translocon. Our study reveals that the C-terminal TMs (cTMs) of numerous MPs across various organisms bypass this cotranslational route, necessitating an alternative posttranslational insertion strategy. We demonstrate that evolution has refined the hydrophilicity and length of the C-terminal tails of these proteins to optimize cTM insertion. Alterations in the C-tail sequence disrupt cTM insertion in both *E. coli* and human, leading to protein defects, loss of function, and genetic diseases. In *E. coli*, we identify YidC, a member of the widespread Oxa1 family, as the insertase facilitating cTMs insertion, with C-tail mutations disrupting the productive interaction of cTMs with YidC. Thus, MP sequences are fine-tuned for effective collaboration with the cellular biogenesis machinery, ensuring proper membrane protein folding.

Membrane proteins (MPs) perform a variety of essential cellular functions and constitute at least a quarter of every proteome[1]. The largest group of MPs, called multispanning MPs, consists of proteins with multiple transmembrane helices (TMs) that span the membrane. A crucial step in the biogenesis and folding of these proteins is the correct insertion of their TMs into the membrane[2]. Inaccurate TM insertion can lead to protein misfolding, loss of function, and in some cases, disease[3].

The insertion of the TMs of multispanning MPs typically occurs co-translationally at the endoplasmic reticulum (ER) membrane in eukaryotes and plasma membrane in prokaryotes and is centered around the universally conserved Sec translocon[4]. In bacteria, the translating ribosome docks on the SecYEG translocon, which inserts the TMs as they emerge out of the ribosome[5,6]. The Oxa1 family insertase YidC and the auxiliary SecDF-YajC complex can cooperate with the SecYEG translocon to facilitate insertion[7,8]. The insertion of mutispanning proteins in eukaryotes is more complex, and was recently shown to involve several additional complexes in the vicinity of Sec61, the eukaryotic SecYEG homolog[9–11]. However,

the fundamental principles of cotranslational insertion are conserved across organisms. Importantly, when each TM finishes its synthesis, it still resides inside the ribosome exit tunnel, inaccessible to the insertion machinery. Geometric considerations, along with experimental studies suggest that the translocon inserts a TM only once an additional stretch of about 45 amino acids has been synthesized C-terminally to the TM (Fig. 1a)[12,13]. This immediately suggests a hurdle – if a TM is too close to the C-terminus and not followed by at least 45 amino acids, it will not be able to insert cotranslationally. Indeed, recent studies have highlighted that post-translational insertion of C-terminal TMs (cTMs) can be inefficient, posing a challenge to biogenesis in yeast and vertebrates[14,15]. Extending a protein's C-tail, either by evolution[14] or artificially[14,15], can allow co-translational insertion and improve biogenesis. However, the C-tails of MPs are often short, necessitating post-translational insertion.

The challenging insertion of cTMs has received much attention in the case of proteins that harbor a single TM close to the C-terminus. Such proteins, called Tail-Anchored proteins, complete their

[1]Department of Biomolecular Sciences, Weizmann Institute of Science, Rehovot, Israel. [2]These authors contributed equally: Ilya A. Kalinin, Hadas Peled-Zehavi. ✉e-mail: nir.fluman@weizmann.ac.il

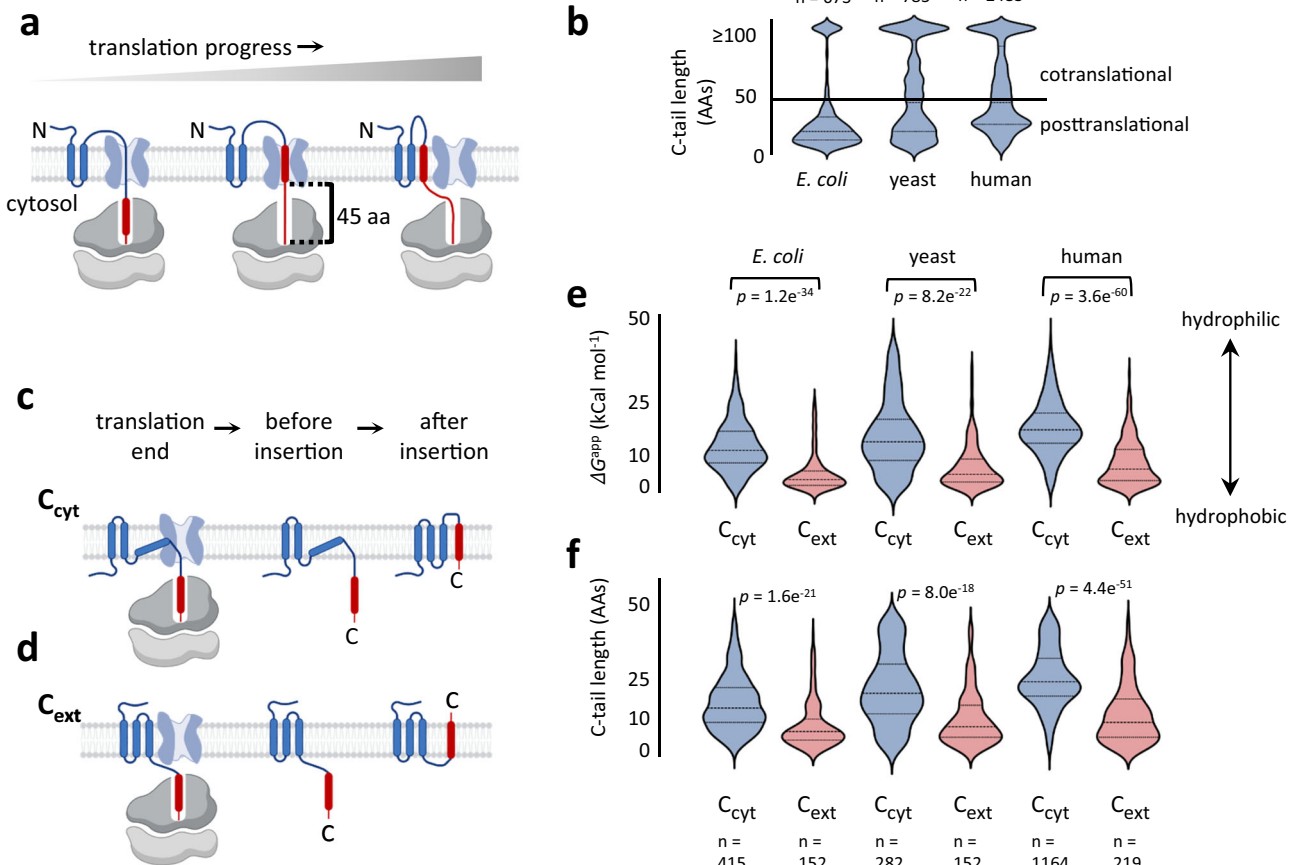

**Fig. 1 | The insertion route of cTMs correlates with the properties of their C-tails. a** Illustration of cotranslational insertion, in which the inserted cTM (red) is accessible to the translocon (light blue) only after 45 amino acids have been synthesized following the TM. **b** The distribution of C-tail length among proteins from different organisms suggests that many proteins depend on posttranslational cTM insertion. **c** and **d** Illustration of the events leading to posttranslational insertion of cTM, in $C_{cyt}$ (**c**) and $C_{ext}$ (**d**) proteins. The events include the end of translation and

posttranslational topological rearrangements. $C_{cyt}$ proteins translocate the penultimate loop, whereas $C_{ext}$ proteins translocate the C-tail. The hydrophilicity (**e**) and length (**f**) of posttranslational C-tails are dramatically different between $C_{cyt}$ and $C_{ext}$ MPs across organisms. $C_{ext}$ proteins have shorter and less hydrophilic C-tails. *p*-values were generated using a two-sided Mann–Whitney test. The *n* values given in (**f**) apply also to (**e**). **a**, **c**, **d** created in BioRender. Kalinin, I. (2023) BioRender.com/g43a126.

translation in the cytosol and must be chaperoned, targeted, and inserted into the membrane post-translationally[16,17]. Several cellular pathways mediate Tail-Anchored protein biogenesis, all converging at post-translational insertases distinct from the Sec translocon. Insertases of Tail-Anchored proteins identified to date include the ER membrane protein complex (EMC) and the guided entry of tail-anchored proteins (GET) complex in the ER membrane, Mtch2 and Mim1 in the mitochondrial membrane, and likely YidC in bacteria[18–23]. However, in contrast to the level of detail we have for Tail-Anchored proteins, we know relatively little about the insertion of cTMs in multispanning proteins. In *Saccharomyces cerevisiae*, the Sec62-Sec63 complex was suggested to play a role in cTM insertion, but this complex is absent from bacteria[24].

Multispanning MPs occupy a vastly different state at the end of translation compared to Tail-Anchored proteins, presenting a somewhat distinct challenge. These proteins are efficiently targeted and anchored to the membrane by their more N-terminal TMs, which insert cotranslationally. Moreover, until translation ends, they are stably docked to the translocon by the ribosome-translocon junction. Their N-terminal, inserted segments may interact with the translocon and its auxiliary subunits. At the moment translation ends, the detachment of the translating proteins from the ribosome may allow them to diffuse away from the translocon. Despite this freedom, studies suggest that multispanning proteins might dwell in the translocon vicinity even after translation[15,25]. However, how common this phenomenon is, and

whether the Sec translocon plays a direct role in cTM insertion was not clear.

Here we set out to elucidate the mechanism of posttranslational insertion of cTMs. We show that the sequences around cTMs, and particularly the hydrophilicity of the C-terminal tails, are constrained by evolution across organisms to optimize insertion. We reveal that for *Escherichia coli* proteins having an extracytosolic C-terminus, a C-tail of low hydrophilicity allows the cTMs to get inserted by the YidC insertase. Increasing the hydrophilicity of the C-terminal tail impairs cTM insertion, leading to protein misfolding and loss of function. A study published during the preparation of this manuscript revealed that cTM insertion in human cells is mediated by EMC, a distant homolog of YidC[26], suggesting that the mechanism of cTM insertion is ancient and conserved. In accordance, we identify disease-causing mutations in human MPs that increase C-tail hydrophilicity, causing misinsertion and mistrafficking consistent with misfolding. Our results indicate that cTMs are inserted by a mechanism distinct from the Sec translocon, and suggest that membrane protein sequences have evolved to optimize their biogenesis through the cellular insertion machinery.

## Results

### Membrane protein C-termini display distinct sequence constraints

Recent studies suggest that C-terminal TMs (cTMs) can insert into the membrane either co- or post-translationally. The main factor

determining the route taken by a protein is the length of the C-terminal tail (C-tail) following the cTM. C-tails shorter than ~45 residues will likely require posttranslational insertion, whereas longer C-tails may allow cotranslational insertion and were shown to improve biogenesis efficiency in some cases[12,13,15,27].

We first asked how many multispanning MPs depend on each route across organisms. To this end, we predicted the topologies of the membrane proteomes of *E. coli*, yeast (*S. cerevisiae*), and human. To improve the prediction accuracy, we combined data from several topology and structure prediction methods (see methods). We then focused on translocon-inserted proteins, by excluding proteins with TMs fewer than 3, and excluding mitochondrial and peroxisomal proteins from the eukaryotic datasets. The resulting dataset contains 673, 785, and 2483 multispanning proteins from *E. coli*, yeast and human, respectively. The distribution of C-tail lengths among these proteins indicates that many membrane proteins have C-tail shorter than 45 residues (Fig. 1b). While in *E. coli* these are the majority (567 proteins), in humans, they are roughly a half (1385 proteins). Thus, post-translational insertion of the cTM is a pervasive necessity during biogenesis.

Since cTM insertion might be challenging, we hypothesized that cTMs may have evolved sequence adaptations to optimize their post-translational insertion. We first focused on the hydrophobicity of the cTM, since hydrophobicity is a major driving force for insertion. The hydrophobicity of posttranslationally inserting cTMs was only slightly different from other TMs. Furthermore, the differences were not consistent across organisms, with cTMs having slightly increased hydrophobicity than other TMs in *E. coli* and decreased hydrophobicity in human (Supplementary Fig. 1a). Similar, slight and organism-specific differences were observed when comparing cTMs that require post- or co-translational insertion (Supplementary Fig. 1b). Thus, we see no evidence for evolution acting on cTM hydrophobicity to optimize insertion.

To explore other sequence adaptations, we focused on another main barrier for insertion—the translocation of the TM's flanking loop/tail across the membrane. The Sec translocon enables the translocation of practically any loop including large polypeptide stretches. However, other insertases struggle with translocating very hydrophilic or long polypeptides[2,28]. Depending on the cTM orientation, the translocated loop may precede or follow the TM. In multispanning proteins that possess a cytosolic C-terminus ($C_{cyt}$), the C-tail remains in the cytosol and the loop preceding the cTM will have to get translocated (Fig. 1c). In contrast, in proteins having an extracytosolic, $C_{ext}$ topology, the C-tail gets translocated (Fig. 1d). We focused on proteins that require posttranslational insertion, possessing C-tails shorter than 45 residues. Our analysis revealed significant differences in the hydrophilicity of their TM-flanking regions, especially in the C-tail following the cTM. In $C_{cyt}$ proteins, the C-tails tend to be longer and more hydrophilic, while in $C_{ext}$ proteins, C-tails are significantly shorter and less hydrophilic (Fig. 1e, f, Supplementary Fig. 2a). Since we calculated tail hydrophilicity by summing the $\Delta G^{app}$ values of all tail residues, much of the tail hydrophilicity is explained by the mere length of the tail. However, the differences in the tail hydrophilicity were sharper than the length differences, suggesting that hydrophilicity is the main adaptive determinant (Fig. 1e, f). Notably, cytosolic loops tend to be longer and more hydrophilic than extracytosolic ones, even when they do not flank the cTM (Supplementary Fig 2a). However, this trend is dramatically sharper in the C-tail, as compared to other loop locations or to the N-terminal tail (Supplementary Fig. 2a). Collectively, these observations suggest that the C-tails are under unique evolutionary pressure, which keeps them hydrophobic or hydrophilic depending on the tail location.

Based on these results, we hypothesized that the hydrophilicity of the C-tail facilitates the correct orientation of cTMs in membrane

proteins. In $C_{cyt}$ proteins, a hydrophilic C-tail may present a barrier preventing its erroneous translocation across the membrane, thereby anchoring the C-tail to the cytosol (Fig. 1c, Supplementary Fig. 2b). By contrast, in $C_{ext}$ proteins, a short and relatively hydrophobic C-tail may enable posttranslational insertion, by presenting a lower barrier for translocation (Fig. 1d).

## Hydrophilic C-tails perturb the insertion of cTM in *E coli*

The distinct sequence attributes of the C-tails suggest an evolutionary constraint, but is this constraint related to cTM insertion? To study this experimentally, we focused on $C_{ext}$ proteins. We chose two *E. coli* proteins with confident topology predictions to study experimentally: UbiA and RcnA, having C-tail sequences 'WHF' and 'R', respectively. The two proteins are structurally unrelated; UbiA is a 4-hydroxybenzoate octaprenyltransferase and has 9 TM, while RcnA is a nickel/cobalt efflux pump with 6 TMs.

We studied their insertion in vivo by a Cys accessibility assay[29] as illustrated in Fig. 2a. Briefly, if the cTM is inserted, the C-tail will translocate to the periplasm. The degree of periplasmic localization of the C-tail is quantified from the relative labeling of a cysteine engineered at the C-tail by AMS (4-Acetamido-4′-Maleimidylstilbene-2,2′-Disulfonic Acid), a Cys-reactive maleimide that labels only periplasmic cysteines as it cannot cross the plasma membrane. As controls, a membrane-permeable agent, N-ethylmaleimide (NEM), labeling both cytosolic and periplasmic cysteines is included, as well as a mock treatment labeling no cysteines. To quantify AMS labeling, the membrane is disrupted, and an additional agent, maleimide-polyethyleneglycol (mal-PEG) is used to label the remaining free cysteines that were not blocked by AMS or NEM. The level of PEGylation blocking by AMS, therefore, reports on the level of periplasmic accessibility.

To simplify the assay's interpretation, we generated cysteine-less variants of these proteins, harboring a single Cys at the C-tail. The C-tails of both UbiA and RcnA were labeled by AMS, blocking the tail's PEGylation (Fig. 2c–e, C-tail extension: 'C'). The accessibility of the C-tails to the periplasmic AMS suggests that the cTM is well inserted. We then tested if the length or hydrophobicity of the C-tail play a role in insertion. The tails of UbiA and RcnA were extended by one to four asparagines (Fig. 2b), increasing their hydrophilicity and length gradually. $C_{ext}$ proteins in *E. coli* typically do not exceed a hydrophilicity of $\Delta G^{app} \approx 5$–6 in their C-tails (Fig. 1e, Supplementary Fig. 2), as calculated by the Biological Hydrophobicity scale[30]. Remarkably, as soon as the asparagine additions to the C-tails crossed the hydrophilicity threshold, the insertion of the cTMs was abolished (Fig. 2c–e). To examine whether length or hydrophilicity affected insertion, we compared the effect of four asparagines with two other four-residue extensions. Extending the C-tail with an HSDS peptide, having similar hydrophilicity to 4xN, similarly abolished insertion. By contrast, a less hydrophilic GSGS extension maintained a wild-type level of insertion (Fig. 2c–e). Thus, sequence hydrophilicity, rather than length, poses the main barrier for insertion. Finally, we investigated the effect of an opsin tag, by appending the sequence GPNFYVPFSNKTG to the C-tail. This tag is a 13-residue-long hydrophilic tag that undergoes glycosylation in eukaryotes and is often used to study MP insertion in the ER membrane[31]. While it is assumed that the Opsin tag does not affect MP insertion, our results show that it abolished insertion in *E. coli* (Fig. 2c–e).

We verified our insertion measurements by testing the effect of the C-terminal extensions on protein function. Since accurate TM insertion is crucial for protein folding and function, we reasoned that mutations perturbing cTM insertion should impair functionality. RcnA is a $Ni^{2+}/Co^{2+}$ exporter, conferring resistance to *E. coli* against these heavy metals. Indeed, extensions to the C-tail that perturbed insertion impaired the ability of mutants to support bacterial growth on $Co^{2+}$-rich medium (Fig. 2f). Collectively, our results suggest that evolution

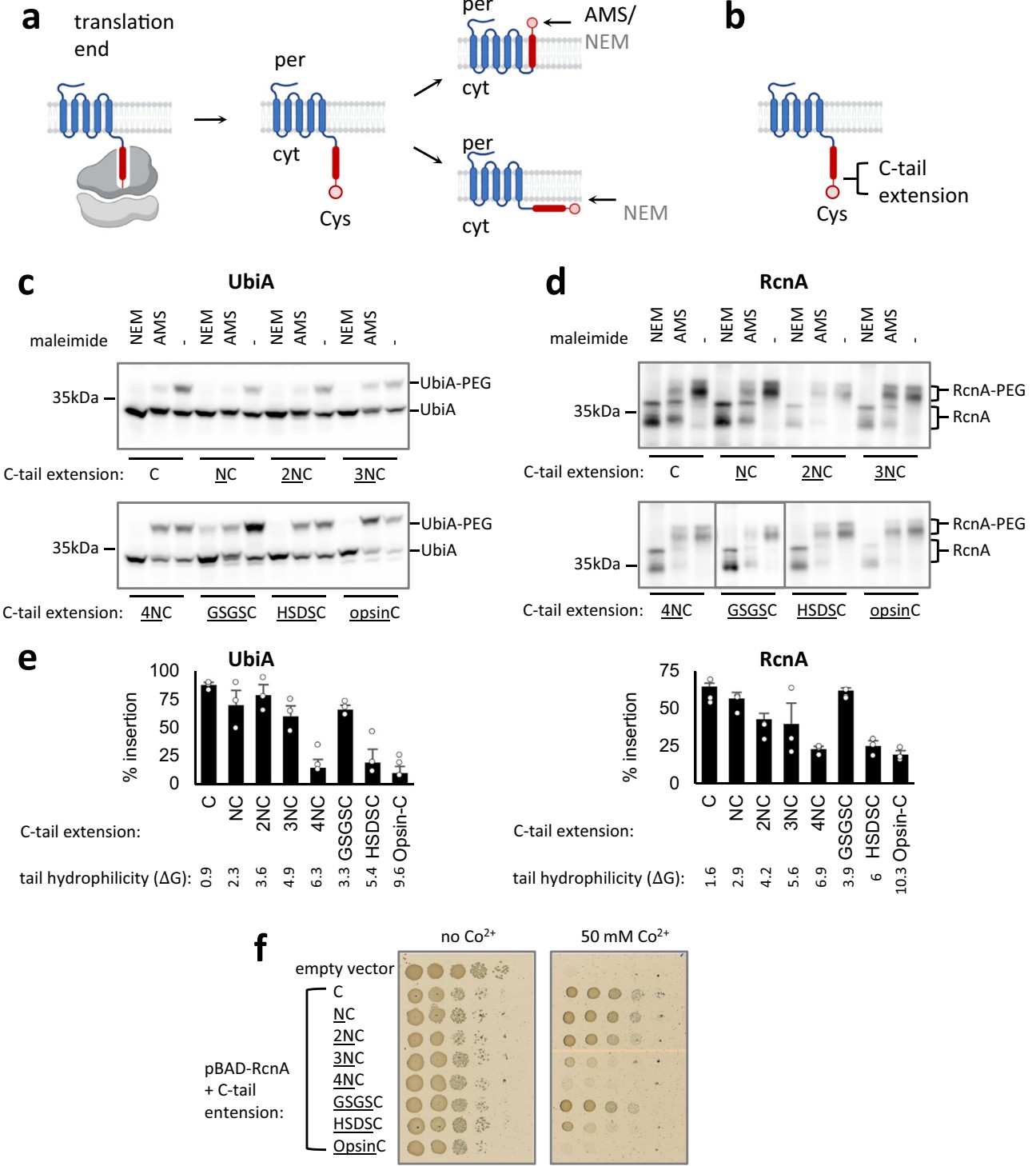

**YidC mediates post-translational cTM insertion in *E. coli***

The observation that hydrophilicity perturbs cTM insertion suggests that the Sec translocon is an unlikely candidate to mediate its insertion since this machinery should be able to handle nearly any loop or tail sequence. We, therefore, set out to identify the alternative insertion mechanism. Beyond the Sec translocon, *E. coli* possesses the YidC insertase, which can insert TMs co- or post-translationally. YidC can work independently or as a part of the holo-translocon complex along with the SecYEG translocon and SecDF-YajC[7,8]. The cytosolic protein

SecA was also shown to participate in targeting and insertion of some MPs[32-34]. Notably, each of these factors may mediate the insertion of TMs other than the cTM, complicating the study of these multi-spanning proteins (see below). We, therefore, first simplified our insertion assay to include only a single TM at the C-terminus. The cTMs and C-tails of RcnA and UbiA, along with a C-terminal Cys, were fused to an N-terminal His$_6$-sfGFP-SUMO protein (Fig. 3a). Since these constructs harbor a single TM at the C-terminus, they cannot insert cotranslationally, similar to Tail-Anchored proteins. Due to technical problems in Cys labeling of the RcnA-cTM construct (Supplementary Fig. 4c), we switched to the construct with GSGS-extended C-terminus, which was properly inserted in the context of the full length protein

**Fig. 2 | Increasing the hydrophilicity of the C-tail in $C_{ext}$ proteins perturbs insertion. a** Measuring cTM insertion in vivo by the accessibility of a C-terminal Cys to the periplasmic reagent AMS. Following translation termination, a Cys located in the C-tail lies in the cytosol (cyt). Upon insertion, it becomes accessible to periplasmic (per) AMS. AMS will therefore block PEGylation of the correctly inserted protein, yet any misinserted protein will not be accessible by AMS and will be efficiently PEGylated even after AMS treatment. NEM labels the C-tail regardless of its insertion status, since it can access both cytosolic and periplasmic Cys. NEM can therefore block the PEGylation of both cytosolic and periplasmic Cys residues. The degree of labeling is quantified from the extent to which AMS blocks Cys PEGylation; For a full explanation of the quantification see Methods. Created in BioRender. Kalinin, I. (2023) BioRender.com/s22f543. **b** Location of C-terminal extensions used to assess how the properties of the C-tail affect insertion. Created in BioRender. Kalinin, I. (2023) BioRender.com/s22f543. **c, d** Periplasmic accessibility assay, measuring the localization of the C-tail by its ability to be labeled by AMS. AMS blocks PEGylation when the protein is properly inserted. Cys PEGylation causes a shift in SDS-PAGE migration, as visualized by Western blotting. Several C-tail extensions perturb cTM insertion and C-tail access to the periplasm. Representative blots are shown for UbiA (**c**) and RcnA (**d**). Note that RcnA displays two bands of slightly different gel migration. **e** Quantifications of the experiments shown in panels (**c**) and (**d**), showing that C-terminal hydrophilicity perturbs cTM insertion. Shown are means ± SEM of at least three biological replicates. **f** $Co^{2+}$-resistance conferred by various RcnA C-tail mutants suggests that mutations perturbing cTM insertion impair RcnA function. A serial 10-fold dilution of ΔRcnA *E. coli* cells harboring an empty vector, or pBAD-RcnA encoding the indicated variant was grown on solid media. Growth In the absence of $Co^{2+}$ is independent of RcnA, whereas growth on $Co^{2+}$-containing media depends on the resistance conferred by active RcnA. Source data are provided as a Source Data file.

(Fig. 2d, e). These chimeric proteins were properly inserted when expressed in *E. coli*, and maintained the dependence on short and hydrophobic C-terminal tails for insertion (Supplementary Fig. 4), suggesting that the insertion mechanism and its sequence dependence were preserved (Fig. 3b, c, 'mock', Supplementary Fig. 4). We then tested the effect of depleting YidC, SecY, SecA or SecD on the insertion of these simplified constructs. We utilized CRISPRi to deplete the expression levels of the factors to 10% or less (Fig. 3d), and confirmed that SecY and SecA were sufficiently depleted to impair their activity (Supplementary Fig. 5). SecY and SecA were dispensable for insertion, confirming that the Sec translocon indeed plays no role in cTM insertion (Fig. 3b, c). By contrast, YidC depletion severely impaired the insertion of the cTMs of RcnA and UbiA, whereas SecD had a minor but consistent effect (Fig. 3b, c). As a control, we tested a similar chimeric construct with a synthetic cTM, termed TM-D, which was shown to be YidC and Sec-independent[35]. Indeed, the depletion of YidC or the Sec components had no effect on TM-D insertion (Supplementary Fig. 6), highlighting the specificity of the effect of YidC on the cTMs of UbiA and RcnA. This suggests that the YidC insertase is the main factor mediating cTM insertion of these multispanning proteins.

We next tested the involvement of the insertion factors in the context of full-length UbiA. Interestingly, YidC, SecY and SecD were all required for cTM insertion in this context (Supplementary Fig. 3a, c, e). We reasoned that the absence of SecY and potentially other translocon components might cause severe misinsertion and misfolding of the entire protein, which may indirectly perturb cTM insertion. Indeed, our analysis shows that SecY, SecD, and YidC are required for the insertion of UbiA TMs other than the cTM (Supplementary Fig. 3b, d, f). We, therefore, postulate that SecY, and potentially SecD and YidC depletion cause a general biogenesis defect in UbiA that may indirectly prevent the cTM from inserting.

To explore the insertion mechanism further, we tested if the His$_6$-sfGFP-SUMO-cTM chimeras physically interact with these insertion factors. The chimeric constructs were briefly expressed in *E. coli* for only 15 min to maximize the analysis of young proteins that may still interact with the insertion machinery. The proteins were then extracted from the membrane using synthetic nanodiscs and purified on metal-affinity chromatography. YidC, SecD and, to some extent, SecY, copurified with the cTM chimeras of both UbiA and RcnA (Fig. 3e). This suggests that YidC interacts with these cTMs, possibly in the context of the holotranslocon. No SecA was detected in the copurified fractions. Surprisingly, however, extending the C-tail by four asparagines had contrasting effects on UbiA and RcnA. For RcnA, the extension abolished its interaction with YidC and SecD, explaining why its insertion was impaired (Fig. 3e). By contrast, for UbiA, the 4N extension intensified the interaction with the factors (Fig. 3e). To examine if the observed interactions occur during insertion, we tested if the proteins dissociate from YidC as they age. To this end, we followed the 15-min induction with a 30-min-long chloramphenicol treatment, which stops the translation of new proteins (Supplementary Fig. 7), allowing us to examine the interactions of older proteins that were given time to diffuse away from YidC. Indeed, after 30 min the UbiA-cTM and RcnA-cTM dissociated from YidC, suggesting that they diffused away in the membrane (Fig. 3f). However, the UbiA-cTM harboring a 4N extension maintained its interaction with YidC even after 30 min with chloramphenicol. The extended interaction suggests that the 4N C-tail extension prevents YidC from efficiently inserting the TM into the membrane, where it will be able to diffuse away.

Collectively, our results suggest that a short and relatively hydrophobic C-tail optimizes the productive interaction of the cTM with YidC, either by regulating the physical interaction or by affecting the insertion rate.

## Disease-causing mutations in C-tails of human MPs perturb cTM insertion

Several elements in the insertion of multispanning proteins are conserved across organisms. The Sec translocon is universally conserved, and both the ER and bacterial inner membranes house Oxa1-family insertases, such as YidC in *E. coli* and EMC, TMCO, and the GET complex in the mammalian ER[36]. Our bioinformatic analysis suggests that the mechanism of cTM insertion might be conserved as well, since similar evolutionary constraints act on the C-tails of proteins from *E. coli* to human, keeping them short and hydrophobic in the case of $C_{ext}$ proteins (Fig. 1e, f). We therefore wondered if increasing the hydrophilicity of C-tails, which perturbs cTM insertion in *E. coli*, may have similar consequences in human.

MP misfolding has been linked to several diseases[3]. We therefore hypothesized that mutations causing cTM misinsertion, and subsequent misfolding, may be pathogenic[14,15]. To test this, we mined the ClinVar database[37] for pathogenic mutations affecting the C-tails of multispanning MPs, concentrating on proteins whose wild-type C-tail length requires posttranslational insertion. We focused on nonstop mutations, where the stop codon is mutated, allowing the translation of an additional polypeptide stretch until the next in-frame stop codon. Notably, such extensions can be detrimental irrespective of insertion. Indeed, while we find pathogenic nonstop mutations in $C_{ext}$ proteins, we also find them in $C_{cyt}$ proteins, though $C_{cyt}$ proteins do not rely on short C-tails for insertion (Fig. 4a). Moreover, in $C_{cyt}$ proteins the pathogenic mutations typically extend the C-tail sufficiently to allow cotranslational rather than posttranslational insertion (Fig. 4a), further suggesting that the mechanism of pathogenicity is likely unrelated to posttranslational insertion. By contrast, the extended tails of five out of seven $C_{ext}$ mutants remain shorter than 45 residues (Fig. 4a, b). These mutants would still rely on posttranslational insertion, but with a substantially more hydrophilic C-tail that may impair insertion.

We next studied the effects of nonstop mutations experimentally. The genes of sodium glucose co-transporter1 (SGLT1), subunit c of the integral V0 domain of vacuolar H$^+$-ATPase (ATP6V0C), and human equilibrative nucleoside transporter 3 (ENT3), along with their nonstop

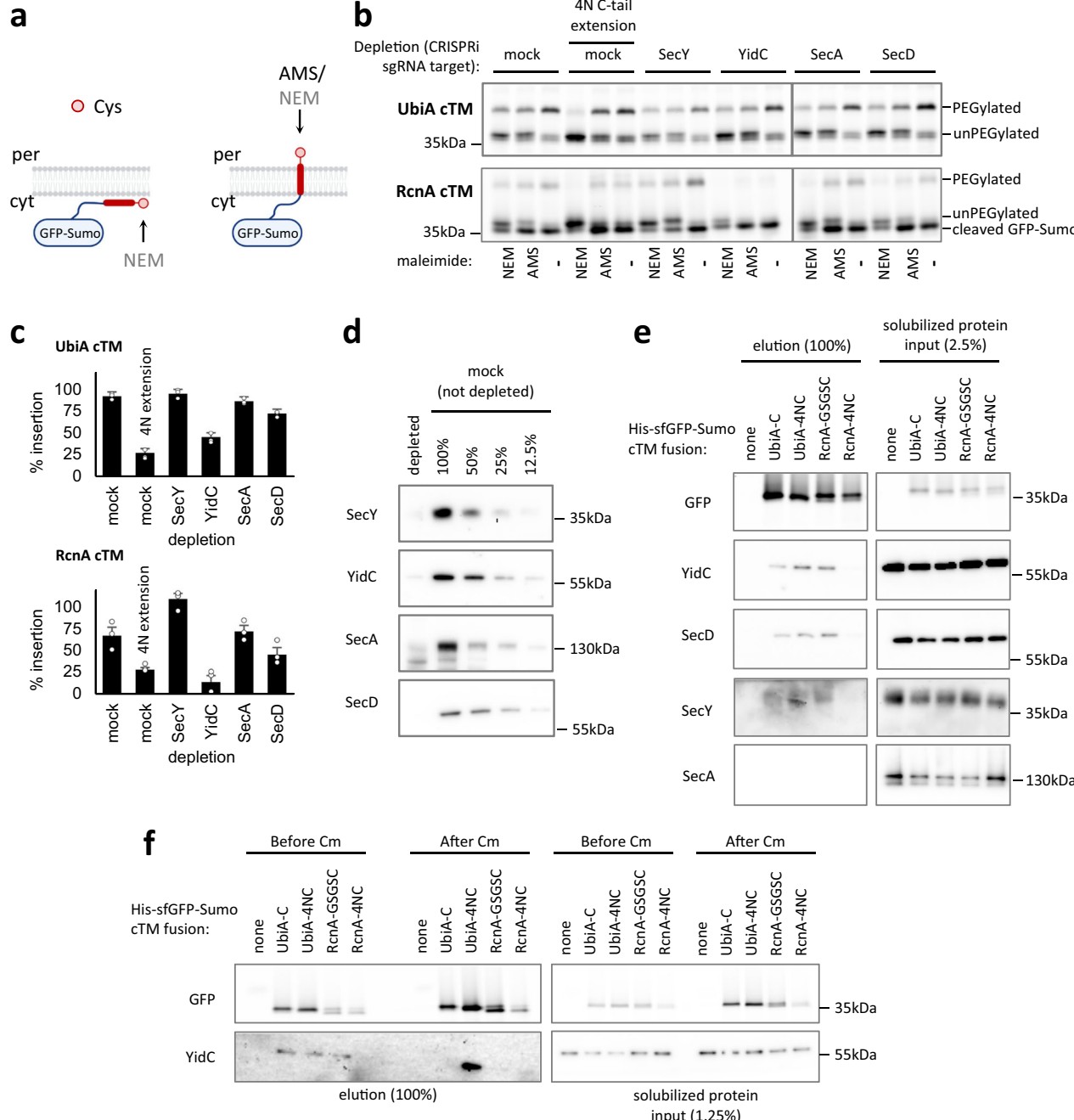

**Fig. 3 | *E. coli* YidC is required for cTM insertion and interacts with cTMs.**
**a** Illustration of the simplified assay for cTM insertion. The cTMs of UbiA or RcnA, along with their C-tails and a C-terminal Cys, were appended to N-terminal $His_6$-GFP-Sumo. If inserted, the C-terminal Cys will be accessible to both NEM and extracellular AMS, whereas an uninserted species will only be accessible to NEM. Created in BioRender. Kalinin, I. (2023) BioRender.com/z82m404. **b** YidC depletion inhibits insertion of UbiA and RcnA cTMs. The in vivo insertion of $His_6$-GFP-Sumo-cTM constructs was assessed by their reaction with AMS and NEM, which blocks PEGylation. CRISPRi was used to deplete the indicated factors using sgRNAs inhibiting their genes. An inert sgRNA was used as control (mock). Note that RcnA-cTM constructs display a significant band consistent with cleaved GFP-Sumo harboring no cTM nor Cys, which does not get PEGylated. **c** Quantification of the level of insertion shows that a 4N extention, YidC depletion, and to a lesser extent SecD depletion impair cTM insertion. Three biological replicates of the experiments shown in (**b**) were quantified. Shown are means ± SEM. **d** Quantitative Western blotting showing that the depletion of various factors reduces their level to below 12%. Total proteins from the depleted cells were run alongside a serial 2-fold

dilution of proteins from undepleted cells and detected with an antibody against the depleted factors. Representative blots of >10 biological repeats are shown. **e** YidC, SecD, and SecY copurify with the chimeric cTM constructs. Synthetic nanodiscs were used to solubilize membranes from whole cells briefly expressing the indicated $His_6$-GFP-Sumo-cTM constructs. Subsequently, the cTM chimeric proteins were purified on $Co^{2+}$ chromatography. The eluted fractions, as well as diluted total protein input samples, were detected for various proteins using the indicated antibodies by Western blotting, or GFP fluorescence. Representative blots of 3 biological repeats are shown. **f** YidC copurifies with newly synthesized chimeric cTM constructs. The cells briefly expressing the cTM were treated with chloramphenicol (Cm) for 30 min, during which time they did not express the indicated His6-GFP-Sumo-cTM constructs. After purification, eluates and input of the solubilized proteins from cells before and after chloramphenicol treatment were examined for the presence of YidC using Western blotting and for GFP using fluorescence. Representative blots of 2 biological repeats are shown. Source data are provided as a Source Data file.

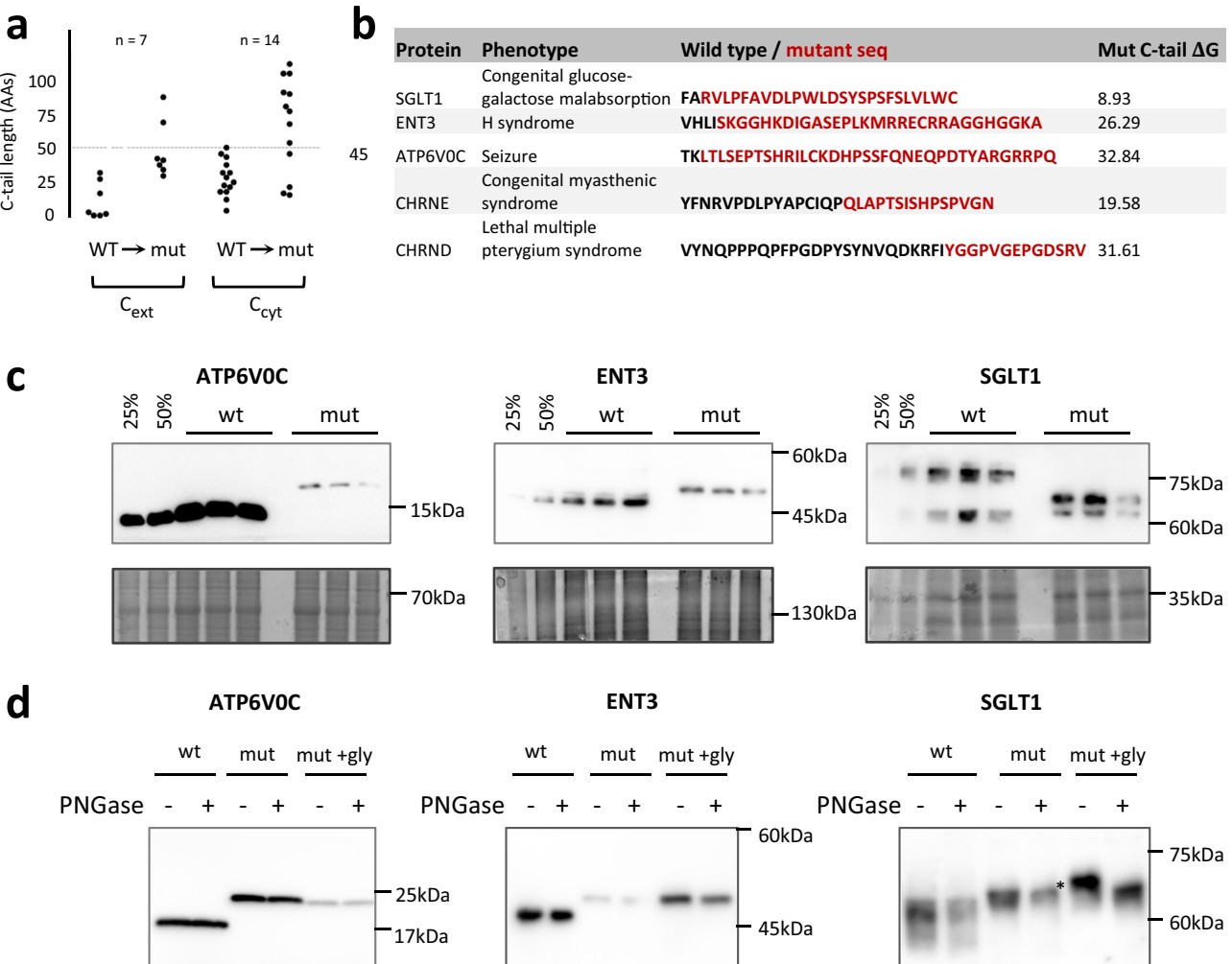

**Fig. 4 | Disease-causing mutations in the C-tails of human MPs impair insertion.** **a** Clinvar-annotated nonstop mutations in posttranslationally-inserting MPs. Pathogenic Mutations were identified in 7 $C_{ext}$ and 14 $C_{cyt}$ multispanning MPs. The length of the C-tail before and after the mutation is shown. Most pathogenic mutations in $C_{ext}$ MPs maintain a C-tail shorter than 45 residues, suggesting that their cTMs would still rely on posttranslational insertion. **b** C-tail sequences of 5 $C_{ext}$ proteins before and after the pathogenic nonstop mutation. **c** Relative expression of ATP6V0C, ENT3, SGLT1 and their nonstop mutants. Lysate from triplicate transient transfections of HEK293, transfected with either the wildtype (wt) or mutant (mut) proteins, were analyzed by immunoblotting with anti-HA antibodies. Equal amounts of total proteins were loaded and Coomassie staining of the gels is shown as a loading control. Serial dilutions of wt sample were used to estimate relative expression. The ATP6V0C mutant displays reduced expression, while the ENT3 and SGLT1 mutants are comparable to wild-type. Representative blots of 3 biological repeats are shown. **d** Glycosylation analysis demonstrates that the C-tails of ATP6V0C and ENT3 nonstop mutants are not translocated to the ER lumen. The endogenous glycosylation site of ENT3 and SGLT1 was eliminated by mutation (see Supplementary Fig. 8) and an NFT glycosylation acceptor site was added to the C-tail of the nonstop mutants of ATP6V0C, ENT3 and SGLT1 (mut + gly). Lysates of HEK293T cells expressing the different mutants were analyzed by immunoblotting with anti-HA antibodies. Where indicated, proteins were deglycosylated with PNGaseF. The addition of the glycosylation site to either ATP6V0C or ENT3 nonstop mutants did not affect the apparent size of the protein (compare mut+gly to mut, left and middle panels), suggesting that their C-tails are not translocated to the ER lumen. In contrast, the addition of glycosylation site to the SGLT1 mutant resulted in an increase in its apparent size that was sensitive to treatment with PNGaseF, suggesting that it is efficiently translocated and glycosylated (right panel). Glycosylated protein band is indicated by asterisk (*). Representative blots of 4 biological repeats are shown. Source data are provided as a Source Data file.

mutants, were transiently expressed in HEK293 cells. The nonstop mutations did not dramatically affect the expression of SGLT1 and ENT3. By contrast, the ATP6V0C mutant showed a drastically lowered expression, hinting at a potential degradation of the misfolded mutant (Fig. 4c). To study membrane insertion we introduced an NFT glycosylation site to the extended C-tails of the ATP6V0C, ENT3 and SGLT1 mutants. Correct insertion of the C-terminus will enable its efficient glycosylation in the ER lumen, resulting in an increase in the apparent size of the protein. SGLT1 and ENT3 harbor an endogenous glycosylation site[38,39]. To simplify interpretation, the asparagine in these sites was mutated to glutamine to eliminate native glycosylation (Supplementary Fig. 8, Q mutants), followed by expression in HEK293T cells to enhance expression. The C-tail of SGLT1 was efficiently glycosylated,

suggesting that the cTM is well inserted and the C-tail successfully enters the ER lumen (Fig. 4d). Thus, the nonstop mutation did not abolish SGLT1 insertion and the mutation's pathogenicity may stem from another reason. By contrast, neither the ATP6V0C nor the ENT3 mutants were glycosylated (Fig. 4d), suggesting that the nonstop mutation prevented the C-tail translocation to the ER lumen. The insertion defect is specific to the extended C-tail, as extracytosolic loops preceding the C-tail in the ATP6V0C and ENT3 mutants are efficiently glycosylated (Supplementary Fig. 9). Thus, the nonstop mutations abolished the cTM insertion of both these proteins.

Finally, we examined the consequences of cTM misinsertion on the biogenesis of the ATP6V0C and ENT3 nonstop mutants. Misfolded membrane proteins typically display a trafficking defect and are

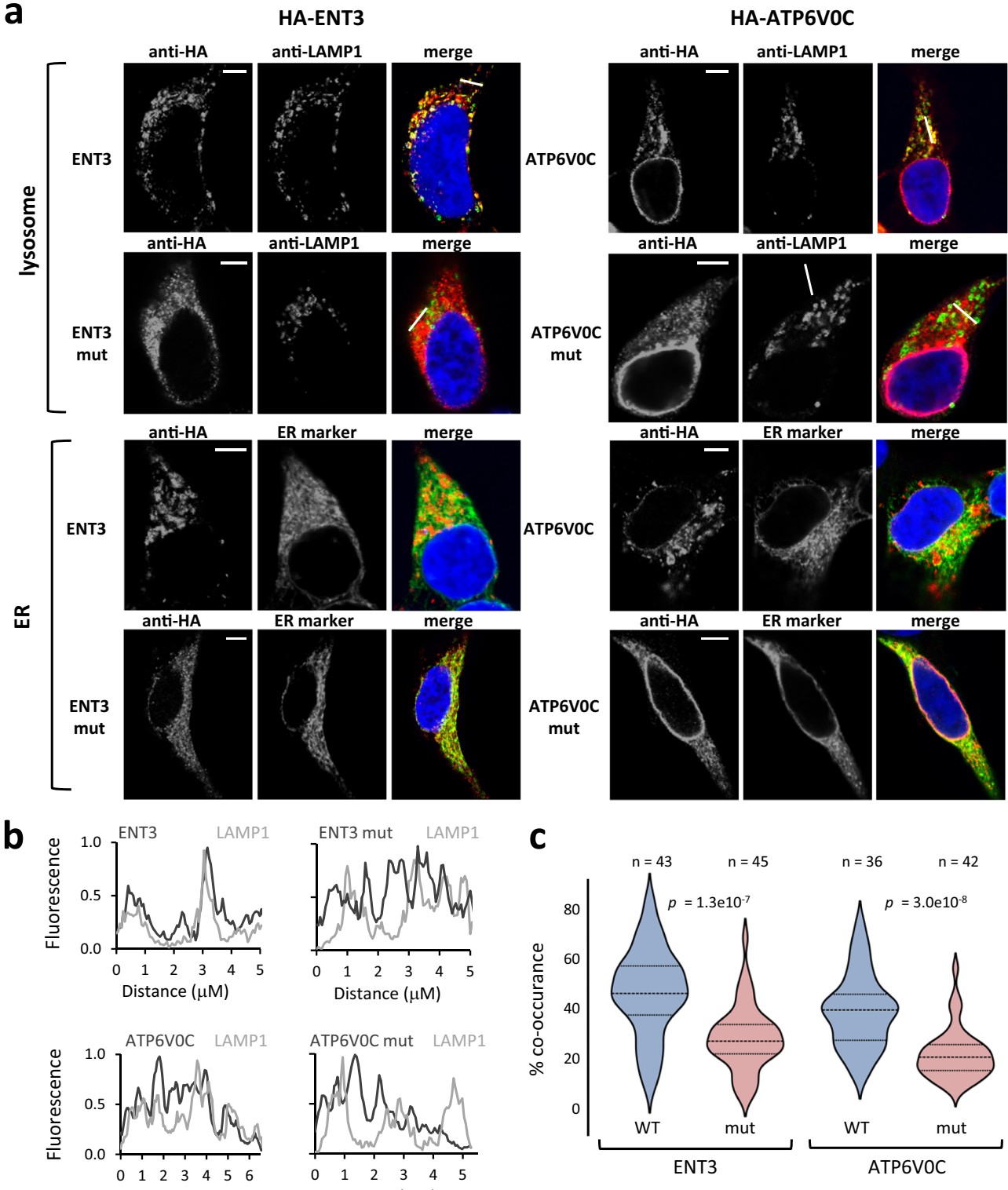

**Fig. 5 | Disease-causing mutations in the C-tails of human MPs display a trafficking defect. a** Immunofluorescence confocal microscopy demonstrates reduced localization of ENT3 and ATP6V0C C-tail mutants to lysosomes. HEK293 cells were transfected with ENT3 and ATP6V0C wildtype or mutant proteins and treated with primary antibodies against HA to label the protein, or LAMP1 to label the lysosome as indicated. A vector expressing ER-mScarletI was used to label the ER (the mScarletI visualization was digitally swapped to green for ease of viewing in the merged images). Colocalization of ENT3 or ATP6V0C (in green) with the organellar markers (in red) results in a yellow signal in the merge panel. DAPY

nuclear staining is shown in blue. Scale bar, 5 mm. **b** Intensity profiles of the fluorescence signals along the white lines indicated in the merge images of the proteins with LAMP1 lysosome marker in (**a**). Fluorescence intensity on the Y axis was normalized to the maximal intensity along the line. Black, ENT3 or ATP6V0C immunofluorescence signal; gray, LAMP1 immunofluorescence signal. The signal of the mutants is not correlated with the lysosomal LAMP1. **c** Quantification of the co-occurrence of the different proteins with lysosomal LAMP1. *p*-values were generated using two-tailed unpaired Student's t-test. Source data are provided as a Source Data file.

retained in the endoplasmic reticulum (ER) by ER quality control. ENT3 is an organellar transporter, partially localized to lysosomes and mitochondria depending on cell type[38,40]. ATP6V0C is a subunit of vacuolar ATPase, a membrane-embedded ATP hydrolysis-driven proton pump that is responsible for acidifying and maintaining the pH of intracellular compartments including lysosomes[41]. Indeed, confocal microscopy shows that wild-type ENT3 and ATP6V0C localize to the lysosome in HEK293 cells (Fig. 5a–c). No significant co-localization of ENT3 with mitochondria was observed in HEK293 cells (Supplementary Fig. 10). In contrast, lysosome localization of both ENT3 and ATP6V0C nonstop mutants is significantly reduced, and their pattern of staining is diffused throughout the cytoplasm, resembling the ER pattern (Fig. 5a–c). These findings are consistent with the expectation that cTM misinsertion would result in mistrafficking due to the misfolding of these mutant proteins.

Collectively, our results suggest that the machinery of posttranslational cTM insertion has a limited capacity to translocate hydrophilic C-tails. This limitation generates an evolutionary constraint, limiting the hydrophilicity of the C-tail in $C_{ext}$ proteins. Mutations that increase the C-tail hydrophilicity perturb this sequence attribute, which in turn abolishes the insertion and folding of the protein, and may lead to disease.

## Discussion

Membrane proteomes encompass a vast array of TMs, ~23,000 TMs in the human proteome and 6000 in *E. coli*. This diversity poses a significant challenge for accurate TM insertion, essential for proteome functionality. Nature has evolved several insertion machineries to handle this diversity, however, the full spectrum of these factors and their client specificities remains incompletely understood[2,3,5,17].

For multispanning MPs, the best-characterized insertion systems are the bacterial membrane's Sec translocon (SecYEG)[6] and its eukaryotic homolog in the ER, Sec61[2]. The Sec translocons can function independently or in concert, forming complexes like the holotranslocon (SecYEG-YidC-SecDF-YajC) in bacteria[8,42,43] or the recently characterized multipass translocon in the ER[10,11]. These complexes allow for the concerted action of the Sec protein-conducting channel, Oxa1-family insertases like YidC in bacteria and TMCO in the ER, and additional factors, all collaborating to efficiently chaperone and insert the multispanning protein[2,7,10,11,34,43–45]. The ER also hosts other Oxa1 family members in the GET and EMC complexes[2,16,36], handling single-spanning proteins[20,22,23] and cotranslational insertion of some N-terminal TMs[46–48].

Our study sheds light on the posttranslational insertion process of a large subset of cTMs in multispanning proteins. We found that in *E. coli*, YidC, an Oxa1-family insertase, mediates the insertion of cTMs that are close to the C-terminus of the protein (Fig. 3). This finding adds a large number of potential clients to YidC, explaining why it is so abundant[49] and essential. It also suggests a potential explanation for the role of YidC in the insertion of proteins like SecE and ATP synthase subunit c[50,51]. Our results align with recent findings that EMC, another Oxa1-family member, performs a similar role in the mammalian ER[26].

YidC, like other Oxa1 family members, does not possess a transmembrane pore capable of translocating long polypeptide stretches. Rather, it contains a hydrophilic groove in the membrane, open from the cytoplasm, that locally thins the membrane, providing a route for TMs to enter while translocating their flanking loops across[52–57]. In line with this mechanism of insertion, the substrates of YidC and other Oxa1 family members typically have short and relatively hydrophobic translocated loops and tails[2,20,28,46,58]. This reliance on Oxa1-family insertases has exerted evolutionary pressure on the C-tails of multispanning MPs, requiring $C_{ext}$ proteins to have short, hydrophobic C-tails to facilitate their translocation. By contrast, the longer, hydrophilic tails of $C_{cyt}$ proteins, might prevent their translocation and anchor them in the cytosol. Thus, by modulating the ability or inability

of the C-tail to translocate, evolution seemingly adapted MP sequences for optimal topogenesis. Similar evolutionary principles may apply to the correct orientation of other TMs or TM pairs that bypass the translocon. This suggests that once translation ends, the Sec translocon can no longer insert most MPs, since this machinery can easily translocate long hydrophilic loops and would not constrain the C-tail sequence.

Comparing the bacterial and human mechanisms reveals notable differences. *E. coli* $C_{ext}$ proteins have uniformly hydrophobic C-tails, whereas human $C_{ext}$ tails exhibit diverse and higher hydrophilicity (Fig. 1e). This suggests a greater capacity of eukaryotic insertases to handle hydrophilic C-tails, as suggested recently for EMC[26]. This increased hydrophilic capacity is also reflected in $C_{cyt}$ proteins, with human proteins requiring a more hydrophilic C-tail to anchor the tail to the cytosol and avoid translocation (Fig. 1e). Yeast proteins populate intermediate hydrophilicity, suggesting that the capacity of Oxa1 members evolved gradually. These conclusions are supported by our experimental results, showing that the cTMs of the bacterial proteins are unable to insert correctly once C-tail hydrophilicity crosses a threshold of $\Delta G^{app} \approx 5-6$. By contrast, a similar increase in SGLT1 C-tail hydrophilicity does not abolish its insertion (Figs. 2e, 4d). Only when the mutations result in a longer and much more hydrophilic C-tail, as is the case for ATP6V0C and ENT3, the threshold hydrophilicity is reached, and C-tail insertion is disturbed (Fig. 4d, Supplementary Fig. 11).

The mechanism of handover of multispanning proteins from the Sec translocon to Oxa1-familiy members is still unclear. In *E coli*, a direct transfer is plausible due to the ability of YidC to function as part of the holotranslocon[8,42]. Indeed, we find that cTMs can interact with additional holotranslocon components, though the functional relevance of this interaction remains unclear (Fig. 3e). The human multipass translocon houses the Oxa1-family member TMCO[10,11], yet whether this protein contributes to the insertion of some cTM remains unknown. By contrast, the human EMC operates independently of Sec61[59,60], suggesting that the protein may need to diffuse away from the translocon to be engaged by EMC. Notably, the N-terminal, already inserted parts of multispanning proteins may interact with EMC cotranslationally, potentially facilitating a swift handover[26,61].

The physiological importance of this insertion mechanism, which likely serves hundreds of MPs in every proteome, is underscored by its implications in disease. Misinsertion of MPs, leading to misfolding and loss of function[14,62], has been implicated in various genetic diseases[15,63,64]. Our findings suggest that human insertases are relatively tolerant to hydrophilic C-tails, making it less likely that single-residue mutations will increase the hydrophilicity sufficiently to impair insertion. However, nonstop mutations pose a greater risk, often resulting in cTM misinsertion and misfolding. Moreover, $C_{ext}$ membrane proteins with naturally hydrophilic C-tails might be more prone to misinsertion and misfolding depending on environmental stresses and the physiological state of the cell. These proteins may be prone to inefficient biogenesis even under normal conditions and undergo extensive degradation, as has been observed for a number of MPs[3]. Our findings also serve to caution against adding tags to these proteins in their C-termini, which may perturb their insertion.

In summary, our research sheds light on the conserved mechanisms of cTM insertion across species, shaped by evolutionary forces and with direct implications for protein misfolding diseases.

## Methods

### Bacterial strains

*E. coli* DH5α was used for plasmid propagation. MG1655(DE3) strain was used for all protein expression experiments. BW25113 ΔRcnA knockout strain (Keio strain JW2093)[65] was used for $Co^{2+}$ resistance assays.

## Bacterial DNA constructs

Gene fragments of cysteine-less UbiA and RcnA were cloned into the pET19b and pBAD24 vectors, respectively, using Gibson assembly. The protein ORFs contained an N-terminal 3× HA tag and a C-terminal cysteine. C-terminal extension variants were generated by either site-directed mutagenesis or Gibson assembly. The UbiA 234C mutant was generated by replacing Glycine 234 in UbiA with cysteine using site-directed mutagenesis. Chimeric GFP proteins were produced by incorporating a SUMO tag followed by the last TM and C-tail of either UbiA or RcnA, along with a C-terminal cysteine, into pBAD/HisB-sfGFP (TIR[STD])[66] (a kind gift from Daniel O. Daley). Details regarding the primers are available in Supplementary Table 1. Guide RNAs (sgRNAs) specific to SecY (tagtAAGCAGTTTGGCAAGTACAG), SecA(tagtCA-GACGTGCACGAAACTCTG), YidC (tagtTCTTTTTCAACGTTATACAG), and SecD (tagtGGAGTCGGTGATATGGTCAC) were integrated into the pFD152 vector[67] using Gibson assembly (pFD152 was a gift from David Bikard; Addgene plasmid # 125546; http://n2t.net/addgene:125546). All constructs were verified by sequencing.

## Bacterial cell growth and protein expression

For full length proteins, MG1655 (DE3) cells harboring plasmids encoding the proteins of interest were grown in LB supplemented with 100 μg/ml ampicillin at 37 °C to mid-log phase (OD$_{600}$ of ~0.5). The cultures were then induced with 0.025 mM IPTG for UbiA variants or 0.01% arabinose for RcnA variants for 30 min induction period, then placed on ice. For CRISPRi experiments, MG1655 (DE3) cells doubly transformed with plasmids encoding the appropriate sgRNAs and GFP-fusion proteins of interest were grown in LB supplemented with 50 μg/ml ampicillin and 40 μg/ml spectinomycin. After 50 min, CAS9 expression was induced by the addition of 25 μg/ml anhydrotetracycline, and the cultures were allowed to grow for an additional 4 h. Protein expression was then induced with 0.02% arabinose for 30 min before the cultures were placed on ice. For pull-down experiments, MG1655 (DE3) cells transformed with plasmids encoding the GFP-chimeric proteins of interest were grown to mid-log phase, then induced with 0.02% arabinose for 15 min before the cultures were placed on ice. In the translation shutoff experiment, where a time gap between protein synthesis and the pulldown was necessary, translation was halted by the addition of 100 μg/mL of chloramphenicol. The cells were then kept growing at 37 °C for an additional 30 min and subsequently placed on ice.

## Cysteine blocking in whole cells

The protocol was adapted from ref. 68. Cells were collected by centrifugation (4 °C, 4700 rpm, 5 min), resuspended in labeling buffer (5 mM MgSO4, 1 mM (tris(2-carboxyethyl)phosphine (TCEP) in PBS)) and divided into three microcentrifuge tubes for blocking with AMS (Setareh Biotech 6508), NEM or labeling buffer. The cells were equilibrated at 37 °C with gentle shaking for 4 min and the sulfhydryl reagents (dissolved freshly in labeling buffer) or labeling buffer were added. Final concentration was 5 mM for AMS and 10 mM for NEM. The cells were allowed to react for 2 or 20 min at 37 °C before quenching for 20 min on ice with dithiothreitol (DTT) at a final concentration of 40 mM. For CRISPRi depletion experiments, a fourth sample of cells was set aside to assess depletion levels of SecY, SecA, YidC or SecD.

## Cell lysis and cysteine PEGylation

For full length proteins, proteins were precipitated by adding trichloroacetic acid (10% final concentration) and incubation for 30 min on ice, followed by centrifugation at (4 °C, 20,000 × g, 15 min). Supernatant was removed and the pellet was washed with ice-cold acetone followed by centrifugation (4 °C, 20,000 × g, 5 min). Acetone was removed and the pellet was completely solubilized in 50 μl of pegylation buffer (100 mM Tris-HCl pH 7.5, 2% SDS, 10% glycerol, 1 mM TCEP, trace amount of bromophenol blue, and protease inhibitor).

13 μl of freshly dissolved 27 mM 5 kDa methoxypolyethylene glycol maleimide (mal-PEG) (Sigma 63187) were added, and the samples were PEGylated with continuous mixing at 30 °C for 2–60 min. The reaction was quenched by the addition of equal volume of stop buffer (2% SDS, 10% glycerol, 40 mM DTT).

For CRISPRi experiments, cells were collected by centrifugation (4 °C, 10,000 × g, 1 min), resuspended in 150 μl of lysozyme buffer (1 mg/ml lysozyme in 150 mM NaCl, 30 mM Tris-HCl, pH 8, 10 mM EDTA, 1 mM TCEP, and protease inhibitor), then frozen at −20 °C overnight. The next day, cells were disrupted by thawing at 25 °C for 5 min followed by shaking at 37 °C for 10 min. Then, 0.9 ml of turbo DNase solution (15 mM MgSO$_4$, 10 μg ml$^{-1}$ turbo DNase (Jena Bioscience, EN-180), 1 mM TCEP and 0.5 mM phenylmethanesulfonyl fluoride) was added and the samples were allowed to shake at 37 °C for 10 min before transferring them to ice. Crude membranes were collected by centrifugation at 4 °C, 20,000 × g for 20 min and the pellet was resuspended in 50 μl of PBS supplemented with 1 mM TCEP and cOmplete protease inhibitor (Roche, 11836170001). Membrane proteins were solubilized and PEGylated by adding 7 μl of 10% β-d-dodecyl maltoside (Anatrace 329370010) and 13 μl of freshly dissolved 27 mM 5 kDa mal-PEG. The samples were incubated with continuous mixing at 30 °C for 2–60 min and the reaction was quenched by mixing with 70 μl of GFP sample buffer (50 mM Tris pH 8, 10% glycerol, 2% SDS, 20 mM DTT, and a trace amount of bromophenol blue).

## Pull-downs

Following induction, 5 OD units of cells (~8.5 mg) were harvested, pelleted, and incubated on ice with lysozyme buffer supplemented with DNase to remove the outer wall. Following a 2 min centrifugation at 10,000 × g, the resulting spheroplasts were washed, resuspended in 80 μl of 5% (w/v) SMALP 140 (Cube Biotech) in purification buffer (20 mM Tris pH 8.0, 500 mM NaCl, 5% Glycerol, 1 mM mercaptoethanol, 5 mM Imidazole, 0.5 mM PMSF), and incubated overnight at 4 °C with shaking (1800 rpm). The sample was then diluted with 900 μl of purification buffer and incubated with shaking for another 30 min, followed by pelleting of unsolubilized debris. The supernatant was incubated with 35 μl of Talon metal affinity resin (TaKaRa) at 4 °C for 3 h. Subsequently, the resin was washed four times with purification buffer, and proteins were eluted with 400 mM imidazole in purification buffer.

## Bacterial protein analysis and Western blotting

All PEGylated samples were resolved on 12% SDS–PAGE gels run with a modified running TRIS-glycine buffer containing only 0.05% SDS. PEGylated full-length proteins were transferred to nitrocellulose membrane and probed with mouse anti-HA antibodies (Sigma-Aldrich, H9658, 1:5000) overnight at 4 °C. PEGylated GFP-chimeras were visualized by Typhoon biomolecular imager (Cytiva) using a 488 nm excitation wavelength and Cy2 filter. For analysis of protein depletion by CriprI, samples were seperated on 4–20% gradient Bis-Tris SDS-PAGE in MES running buffer, then transferred to nitrocellulose. The membrane was probed with polyclonal antibodies against SecY (1:3000), SecA (1:10,000), YidC (1:10,000), SecD (1:2500)[69], and OmpA (1:10,000, Bioorbyt orb862303, S1304). Pull-down GFP-chimera samples were separated on 12% SDS-PAGE gels in Tris-Glycine containing 0.05% SDS, followed by in-gel fluorescence detection. Subsequently, the samples were transferred to nitrocellulose and probed with polyclonal antibodies against SecY, SecA, YidC (1:5000), and SecD as above. ImageQuant™ TL 10.2 analysis software (Cytiva) was used for band quantification. To ensure maximal accuracy of PEGylation level quantification, a calibration curve was constructed for each experiment using a series of twofold dilutions of sample.

## Calculation of the percent of insertion

The percent of insertion is calculated by the percent to which AMS can modify the periplasmic Cys, thereby blocking subsequent PEGylation. In order to calculate it, the percent of PEGylation is quantified by densitometry of SDS-PAGE gels or Western blots, from three samples: an NEM-modified sample (representing the maximal possible blocking of PEGylation, as NEM can modify both cytosolic and periplasmic Cys; the condition controls for inaccessible cysteines which become accessible only after cell lysis and cannot be blocked in whole cells), an AMS-modified sample (representing the blocking emerging from periplasmic Cys only), and an untreated sample (representing the maximal possible PEGylation). Percent insertion is then defined as

$\%insertion = 100 * (1 - \%PEGylation_{AMS} / \%PEGylation_{untreated}) / (1 - \%PEGylation_{NEM} / \%PEGylation_{untreated})$.

## Co²⁺ resistance assay

ΔRcnA knockout strain was transformed with pBAD24 plasmid with and without the different RcnA C-tail mutants. Cells were grown to mid-log on M63 minimal medium supplemented with 0.2% glycerol and 0.4% glucose, and tenfold serial dilutions were prepared with the highest density having an $OD_{600}$ of 0.1. The serial dilution was spotted (3 μL) on supplemented M63 agar–ampicillin plates containing 0.1% arabinose and either 0 or 50 mM $CoCl_2$. The plates were allowed to grow for one night at 37 °C. As mutation of the conserved cysteine187 of RcnA reduces its activity, the complementation assay was done with RcnA C-tail mutants that retain this cysteine.

## Mammalian cell culture and transient transfections

HEK-293 cells (ATCC, CRL-1573) and HEK-293T (ATCC, CRL-3216) cells were subcultured every 3–4 days in Dulbecco's modified Eagle's medium (Thermo Fisher Scientific, 41965-039) supplemented with 10% v/v fetal bovine serum (Sigma Aldrich, F7524), 1% l-glutamine (Biological Industries – Sartorius, 03-020-1B), and 1% Penicillin Streptomycin Solution (Biological Industries – Sartorius, 03-031-1B) under standard conditions (37 °C and 5% $CO_2$ in a humidified incubator). Cells were routinely tested by PCR for the absence of mycoplasma contamination. HEK-293 cells were transiently transfected with constructs using the JetPRIME reagents (Polyplus Transfection 101000015) and HEK-293T cells were transfected using the PolyJet DNA transfection reagent (SignaGen Laboratories SL100688) according to the manufacturers' instructions. Typically, 1.0–1.5 μg DNA was used to transfect cells i n a 6-well plate (0.4 μg DNA for wild type ATP6V0C). Cells were harvested 44–48 h after transfection.

## Mammalian DNA constructs

Gene fragments of ATP6V0C, ENT3 (SLC29A3) and SGLT1 (SLC5A1) and their stop codon mutants (ClinVar variants NM_001694.4(ATP6V0C):c.467A>T (p.Ter156Leu), NM_018344.6 (SLC29A3):c.1427A>G (p.Ter476Trp) and NM_000343.4(SLC5A1):c.1993T>C (p.Ter665Arg)) were cloned into pCDNA3.1-HA (Addgene 128034) using Gibson assembly. Glycosylation site mutagenesis of ENT3 (N84Q) and SGLT1 (N248Q) and addition of an NFT glycosylation site 6–9 amino acids from the C-terminal of all three proteins was done by site-directed mutagenesis PCR. Glycosylation sites in periplasmic loops were generated by adding an opsin tag (MNGTEGPNFYVPFSNKTVD) between amino acids 78 and 79 in the extracytosolic loop preceding the C-tail of ATP6V0C, and between amino acids 335 and 336 (loop 8) or 401 and 402 (loop 10, the extracytosolic loop preceding the C-tail) of ENT3. Details regarding the primers are available in Supplementary Table 1. All constructs were verified by sequencing.

ER-mScarletI[70] (Addgene 137805) was used as an ER organellar marker.

## Mammalian protein extraction, deglycosylation assays and immunoblotting

For expression levels analysis, cells were washed once with ice-cold PBS and then lysed with SDS lysis buffer (2% SDS), 50 mM TRIS-HCl pH6.8, 10% glycerol, turbo nuclease and cOmplete protease inhibitor for 20 min at 37 °C. The lysate was quantified using BCA protein assay (Pierce, 23227) and supplemented with a trace of bromophenol blue and DTT (0.1 M final concentration). Equal amounts of proteins were loaded on gels. For deglycosylation experiments, cells were lysed 30 min on ice in triton lysis buffer (50 mM Tris-HCl pH 8, 150 mM NaCl, 1% triton, 1 mM EDTA and protease inhibitor). The lysate was centrifuged for 15 min at $20,000 \times g$ at 4 °C, and the supernatant was digested for 1 h at 37 °C with PNGase F (New England Biolabs, P0704) according to the manufacturer's instructions. Negative controls were prepared in the same manner without the addition of enzymes. Following digestion, the samples were supplemented with 4× reducing Laemmli buffer.

Cell lysates were analyzed by SDS–PAGE on 12% Tris-glycine gels. SDS–PAGE gels were transferred to a nitrocellulose membrane (for ATP6V0C samples) or PVDF membranes (for ENT3 and SGLT1). Membranes were probed with mouse anti-HA antibodies (Sigma-Aldrich, H9658, 000013287, 1:5000) overnight at 4 °C.

## Immunofluorescence staining

HEK-293 cells were seeded on poly-L-lysine (Sigma Aldrich, P4707) coated cover slips in a 24-well plate and transiently transfected the following day with 50–250 ng DNA. The cells were fixed 44–48 h after transfection. If required, MitoTracker® Deep Red FM (Invitrogen, M22426) at a final concentration of 125 nM was added to the growth media of cells prior to fixation. Following 10 min incubation with MitoTracker®, the cells were washed once, growth media was replaced, and the cells were incubated for another 30 min. The cells were fixed with 4% PFA in PBS for 15 min at room temperature, permeabilized with 0.1% saponine in PBS for 10 min, and blocked in 10% FBS in PBS (blocking buffer) for 1 h. The cells were then incubated with primary antibodies in blocking buffer for 2 h, and with secondary antibodies for 1 h, followed by 10 min incubation with DAPI at 0.5 μg/ml. Slides were mounted in Fluoromount-G (SouthernBiotech Associates, 0100-01). The following antibodies were used: mouse anti-HA (BioLegend, 901501, B350094, 1:750), rabbit anti-HA (Sigma-Aldrich, H6908, 0000160996, 1:750), mouse anti-LAMP1 (DSHB, H4A3-c, 1:200), goat Anti-Mouse IgG H&L Alexa Fluor® 488 (Abcam, 150113, 1004120-69, 1:500) and donkey Anti-Rabbit IgG H&L Alexa Fluor® 568 (ab175692, 1027933-12, 1:600).

## Confocal microscopy and image analysis

The cells were imaged with a DMi8 Leica confocal laser-scanning microscope (Leica Microsystems, Mannheim, Germany), using ×60/1.4 oil-immersion objective. Fluorescence of DAPI, Alexa488, and Alexa568 or mScarlet was acquired using 405 nm, 488 nm and 561 nm lasers, with emission collection width set to 415–460, 500–550, and 570–630, respectively. Fluorescence of MitoTracker® Deep Red FM was acquired by white-light laser using a 641 nm laser line with emission collection width set to 650–750.

The images were analyzed using Fiji software. Background subtraction was performed using control samples treated only with secondary antibodies (for LAMP1) or by using mock transfection controls (for HA-tagged proteins or the ER-mScarletl marker). Thresholds were determined using Otsu's threshold clustering algorithm, followed by pixel- to-pixel measurement of co-occurrence in the region of interest.

## Topology predictions

The entire reviewed proteomes of *E. coli* K-12 (taxon identifier 83333), *S. cerevisiae* (559292) and human (9606) were downloaded from

Uniprot (as of January 2023) and were parsed using custom Python scripts. To analyze the topologies of the proteomes, topologies were downloaded from three sources: CCTOP[71], TMAlphaFold, and PolyPhobius[72]. From TM-AlphaFold, we kept only proteins that passed at least 5 tests. To obtain Polyphobius predictions, we utilized the TOPCONS server. We crossed data from all predictors to check the level of agreement in the number of TMs and the position of their core residues (within a 14-residue window). In cases where the three predictors agree, we labeled the topology as 'high' quality. When disagreements were present, we prioritized topologies where CCTOP and TMAlphaFold agree, followed by agreements between TM-AlphaFold and PolyPhobius. When there was no agreement, or when data were missing, we preferred Polyphobius predictions over CCTOP and the topologies were labeled as 'low' quality.

The membrane topology (TM boundaries and membrane orientation) and biological hydrophobicity ($\Delta G^{app}$ values for TM insertion) of the entire proteome were analyzed by PolyPhiobius and the $\Delta G$-predictor, as implemented in the Topcons webserver[73]. Proteins longer than 10,000 or shorter than 18 residues were removed. The amino acid 'U' (selenocysteine) was substituted by cysteine. To assign the hydrophobicity $\Delta G^{app}$ value for each TM, the minimal $\Delta G^{app}$ given by the $\Delta G$ predictor value within five to six residues of the TM center was kept. Signal peptides were ignored. To analyze loop hydrophobicity, the $\Delta G^{app}$ values of all residues within the loops were summed. The $\Delta G^{app}$ values for loops were taken from the biological hydrophobicity scale[30], corresponding to residues positioned in the middle of the membrane. The topology predictions are provided in Supplementary Data 1. To preferentially analyze co-translationally inserted proteins, proteins containing less than 3 TMs were discarded. In addition, we discarded mitochondrial and peroxisomal proteins. The identities of yeast mitochondrial and peroxisomal proteins were provided as a personal communication from the lab of Maya Schuldiner. Human mitochondrial proteins were taken from MitoCarta3.0 on March 2022[74]. human peroxisomal proteins were from[75].

### Reporting summary

Further information on research design is available in the Nature Portfolio Reporting Summary linked to this article.

## Data availability

Supplementary Data 1 contain the membrane proteins dataset used in this study, including topology predictions and $\Delta G^{app}$ calculations. All data generated in this study are provided in the Supplementary Information/Source Data file. Source data are provided with this paper.

## Code availability

The source code used to produce the topology predictions of the membrane proteomes of *E. coli*, yeast and human (Supplementary Data 1) is available via Zenodo (https://doi.org/10.5281/zenodo.13927842)[76].

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

## Acknowledgements

We thank Ofer Shoshani and Karen Hakeny for invaluable help with establishing human cell culture facility and methods; Yael Alon, Ivan Milenkovic, Natali Muskat and Inna Goliand for their helpful advice on immunofluorescence staining and confocal imaging, Yoav Peleg for consultations on the cloning, and Eitan Bibi for kindly providing SecY, SecA, SecD and YidC antibodies. This research was generously supported by research grants, to N.F., from the ISRAEL SCIENCE FOUNDATION (grants no. 2207/21 and 2208/21), the Kekst Family Institute for Medical Genetics, the Center for New Scientists at the Weizmann Institute of Science, and the Weizmann SABRA - Yeda-Sela - WRC Program, the Estate of Emile Mimran, and The Maurice and Vivienne Wohl Biology Endowment. Illustrations were Created with BioRender.com.

## Author contributions

I.A.K., H.P.-Z., and N.F. designed the study and the experiments. I.A.K. and H.P.-Z. conducted the experiments. I.A.K., A.B.D.B., S.A.T., and Y.W. performed the computational analysis. H.P.-Z. and R.N. performed the image analysis. N.F., I.A.K., and H.P.-Z. conducted the majority of data analysis. N.F. and H.P.-Z. wrote the manuscript. All authors discussed the results and contributed to editing the manuscript. H.P.-Z and N.F. conceived the study. N.F. supervised the work.

## Competing interests

The authors declare no competing interests.
