## [Transparent Peer Review file · Nature Communications]

Features of membrane protein sequence direct post-translational insertion

Corresponding Author: Dr Nir Fluman

Version 0:

Reviewer comments:

Reviewer #1

(Remarks to the Author)

Main Comments

This paper, submitted to Nature Communication by Kalinin et al, provides data to support the idea that multispanning proteins in *E. coli* and eukaryotes that have C-terminal TM segments that bypass the co-translational route employ Oxa1 family of proteins to insert their C-terminal tails across the membrane. Interestingly, they show that these proteins in *E. coli* typically have short C-tails with low hydrophilicity. They then study the requirements for insertion of the C-terminal tail employing UbiA and RcnA that span the membrane 9 and 6 times, respectively. Translocation of the C-terminus is assayed by using AMS (a cysteine-specific membrane impermeable reagent) to modify a single cysteine introduced into the tail region. When the hydrophilicity is increased, the C-terminal tails are inhibited or blocked in translocation. These results are quite clear. The authors explored the insertase requirements using artificial single TM constructs of UbiA or RcnA (comprised of GFP-Sumo fused upstream to the C-terminal TM segment) and CRISPRi to deplete either YidC, SecY, SecA, or SecD. The results suggest that YidC, the Oxa1 bacterial homolog, may be involved in inserting the C-terminal tail. The studies in Fig. 4 and 5 investigate disease-causing mutations of human membrane proteins that extend the C-tail, and the authors propose that these mutations perturb insertion of the C-tail into the ER lumen, causing the misinserted proteins to be retained in the ER.

Overall, the results showing that an Oxa1 homolog is involved with multispanning membrane proteins in the post-translational insertion of the C-terminal tail are interesting and new, but the results are still preliminary. The studies in Figures 1 and 2 are solid and support the authors' conclusions. However, there are problems with the studies in Figure 3 since they switched from studies of the multispanning proteins, UbiA and RcnA, to artificial constructs containing the C-terminal TM segments and C-tails of UbiA and RcnA, along with a C-terminal Cys, fused to an N-terminal His6-sfGFP-SUMO protein. While it may be easier to study single TM segment proteins, it is important to perform the depletion experiments with the original parent constructs studied in Figure 2. Also, as it stands, the studies in Figures 4 and 5 don't really fit with the bacterial studies, because the authors did not show that the studied eukaryotic proteins (ATP6VOC, ENT3 and SGLT1) use an Oxa1 homolog for insertion, although they expect they do. Additionally, there are some technical issues (see below) that need to be addressed to make sure the authors' conclusions are correct. Therefore, while the main message is very interesting and point to an important new role for Oxa1 family members, the conclusions in the submitted paper are still too preliminary to support their conclusions fully.

Other Comments

- 1.State explicitly how many of the Fig. 1b multispanning membrane proteins in *E. coli*, yeast and humans have posttranslational C-tails with the carboxy terminus located away from the cytoplasm (Cext) (i.e. must translocate across the membrane).
- 2.Please provide the length and the amino acid sequence of the C-tail of UbiA and RcnA.
- 3.For the delta G values to determine the hydrophilicity of the posttranslational C tails in Fig. 1e, Fig. 2e and Fig. 4b, why

was the contribution of the peptide bonds not included since this region is translocated across the membrane?

4. Please provide the equation that was used to calculate the Percent Insertion (translocation) of the C-tails in Fig. 2e and Fig. 3c. By eye, the AMS/PEGylation data to assay for translocation of the C-tails (Fig 3b) does not seem to agree with the quantification of insertion results (Fig. 3c). For example, the AMS/PEG studies with the UbiA cTM construct when either SecY or YidC was depleted look quite similar (Fig. 3b). But the quantification of % insertion (Fig. 3c) is quite different. It is not clear how insertion can be accurately determined for the RcnA cTM constructs since so little of the construct is capable of being modified by PEGylation even when no AMS is added (see Fig. 3b).

5. The authors state that the C-terminal tail translocates slowly for cTM-4NC because they see more AMS modification at 20 min compared to 2 min. This result may not have anything to do with insertion rate but simply be due to the fact AMS modification of the cysteine in the C-tail was not complete at 2 mins, and needed more time. Actually, it is not clear to me how you can look at the rate of insertion by looking at chemical amounts of the protein by western blotting. Typically, one looks at the translocation rate of newly synthesized proteins by doing a pulse chase experiment (radiolabel cells with S35 methionine (for example) for 2 min with no chase or with a 20 min chase with non radioactive methionine).

6. It is not clear what the results of the co-purification of the UbiA and RcnA His tagged sfGFP-Sumo cTM variants with YidC, SecD, or SecY (or lack of) mean since the authors did not measure the interaction during insertion. Typically, you would want to probe the interactions of the substrate with the translocase/insertase as insertion is occurring, not after the membrane insertion process nor with chemical amounts of the proteins extracted from the membrane.

7. The authors might consider studying the insertion of newly synthesized UbiA and RcnA using a short radioactive pulse of S35-methionine and then immunoprecipitate the protein with antibody against the HA tag. The YidC, SecD and SecY depletion studies can be carried out with newly synthesized proteins. The AMS/NEM/PEG method could be used with newly synthesized [35S]-labeled construct to show the translocation of the C-terminal tag is inhibited. I would also recommend as a control that the authors show that the co-translational inserted loop preceding the C-terminal tail is blocked by SecY depletion. To do this, they can make another construct and add a cysteine to this loop, and examine its translocation using the AMS accessibility assay.

8. It is necessary in the studies described in Fig. 4 to show the C-tail of the parent wild-type ATP6VOC, ENT3, and SGLT1 is translocated. The authors should show the loop preceding the C-tail of the ATP6VOC, ENT3, and SGLT1 mutants is translocated using the PNGase. A glycosylation site can be introduced in this loop.

9. Why are there two bands for RcnA in Fig. 2d?

Reviewer #2

(Remarks to the Author)

The manuscript by Kalinin et al identifies a key problem in polytopic membrane protein biogenesis, that proteins with short soluble C-termini must integrate their last transmembrane domain post-translationally. This is distinct from each of the preceding TMs. They take a broad approach to studying this problem, using bioinformatics and experimental studies that the hydrophobicity of these proteins has been finely tuned to optimize insertion. Furthermore, they demonstrate that in bacteria the insertion of these proteins is dependent on the insertase YidC. This is an elegant study that addresses the evolution of membrane proteins and provides a new model for how these proteins are synthesized. They demonstrate that this knowledge has implications for disease and protein expression. I have no major concerns about this work.

Minor concerns:

Introduction: Mim1 should be capitalized.

Results:

Pg. 4 – It would be useful to have more of an explanation for the choice of UbiA and RcnA. Are they representative?

Pg. 6 – For the results in Fig 4, using terms like most for 5 of 7, while correct, to me is misleading. You should just use 5 of 7, be explicit when the numbers are small.

Discussion:

These results will have implications for the expression of membrane proteins. It would be nice to include some thoughts towards that.

Figures:

1e&f. have similar scales (0-50). Just make sure that is correct. They are measuring different things.

2a. For clarity, label which (top/correct insertion or bottom/no insertion) should be PEGylated.

2c. For the GSGSC extension, why is there PEG in the NEM control.

4a. In this case, I find the violin plots misleading. For such small numbers, the scatter clustering is sufficient to convey the results.

4c. What are the multiple bands for SGLT1?

Version 1:

Reviewer comments:

Reviewer #1

(Remarks to the Author)

The reviewers have satisfactorily addressed my main concerns in the previous round of review. I find the paper acceptable.

Reviewer #2

(Remarks to the Author)

The paper has been improved in this revision. I have no further concerns.

Reviewer #2 (Remarks to the Author)

Main Comments

This paper, submitted to Nature Communication by Kalinin et al, provides data to support the idea that multispinning proteins in E. coli and eukaryotes that have C-terminal TM segments that bypass the co-translational route employ Oxa1 family of proteins to insert their C-terminal tails across the membrane. Interestingly, they show that these proteins in E. coli typically have short C-tails with low hydrophilicity. They then study the requirements for insertion of the C-terminal tail employing UbiA and RcnA that span the membrane 9 and 6 times, respectively. Translocation of the C-terminus is assayed by using AMS (a cysteine-specific membrane impermeable reagent) to modify a single cysteine introduced into the tail region. When the hydrophilicity is increased, the C-terminal tails are inhibited or blocked in translocation. These results are quite clear. The authors explored the insertase requirements using artificial single TM constructs of UbiA or RcnA (comprised of GFP-Sumo fused upstream to the C-terminal TM segment) and CRISPRi to deplete either YidC, SecY, SecA, or SecD. The results suggest that YidC, the Oxa1 bacterial homolog, may be involved in inserting the C-terminal tail. The studies in Fig. 4 and 5 investigate disease-causing mutations of human membrane proteins that extend the C-tail, and the authors propose that these mutations perturb insertion of the C-tail into the ER lumen, causing the misinserted proteins to be retained in the ER.

> We thank the reviewer for their insights and criticism, which helped us greatly enhance our paper. Below please find our answer to each of the points raised. We hope that the reviewer will agree that the paper is much improved in its current form.

Overall, the results showing that an Oxa1 homolog is involved with multispinning membrane proteins in the post-translational insertion of the C-terminal tail are interesting and new, but the results are still preliminary. The studies in Figures 1 and 2 are solid and support the authors' conclusions. However, there are problems with the studies in Figure 3 since they switched from studies of the multispinning proteins, UbiA and RcnA, to artificial constructs containing the C-terminal TM segments and C-tails of UbiA and RcnA, along with a C-terminal Cys, fused to an N-terminal His6-sfGFP-SUMO protein. While it may be easier to study single TM segment proteins, it is important to perform the depletion experiments with the original parent constructs studied in Figure 2.

> We agree with the reviewer that it would be ideal to study the insertion of the C-terminal TM in the context of the full protein. However, this is experimentally nearly unachievable, because the insertion factors, such as SecY and potentially YidC, surely affect the insertion of the more N-terminal TMs. Depletion of these factors is likely to cause a massive biogenesis, insertion, and folding defect for the entire protein, complicating the interpretations. For example, if SecY depletion impairs cTM insertion,

this can be caused by problems in targeting or insertion of the more N-terminal TMs. Therefore, to study the involvement of the translocon components or YidC in cTM insertion, it is vital to study them using a construct that is otherwise independent of these factors. While we were not initially sure that a single-spanning membrane protein, harboring only the cTM, would be ideal, our studies (Fig. 3b) revealed that this construct retained the dependence of insertion on C-tail hydrophobicity, suggesting that the mechanistic principles of sequence specificity are preserved in this construct.

To support our approach, we provide two additional experiments. The new Fig. S4 shows a deeper characterization of the C-tail sequence requirements for insertion. We find that the GFP-SUMO-cTM constructs almost completely mirror the sequence requirements for insertion that were observed for the full-length protein. Namely, UbiA-cTM can be inserted efficiently with up to 3xNs, and a 4xN extension abolishes insertion. UbiA-cTM insertion can tolerate a GSGS extension, just like the full length protein. RcnA shows a more gradual reduction of insertion with the addition of Ns, but insertion is abolished by a 4xN or HSDS extension and is unaffected by a GSGS extension. These results are almost identical to those obtained with the full-length protein (Fig. 2c,d), with the exception that UbiA insertion in the GFP-SUMO-cTM construct can tolerate an HSDS extension better than the the full-length construct. These results suggest that the sequence specificity of insertion is well preserved, suggesting that the chimeric single TM constructs are a viable tool to understand what is the molecular mechanism responsible for this sequence requirement.

Another support comes from the new Fig S3, which shows that depletion of each of the insertion factors, YidC, SecY, and SecD abolishes insertion of not only the cTM, but also more N-terminal TMs. We have now added a few sentences that discuss these results, which read: “We next tested the involvement of the insertion factors in the context of full-length UbiA. We next tested the involvement of the insertion factors in the context of full-length UbiA. Interestingly, YidC, SecY and SecD were all required for cTM insertion in this context (Fig. S3a,c,e). We reasoned that the absence of SecY and potentially other translocon components might cause severe misinsertion and misfolding of the entire protein, which may indirectly perturb cTM insertion. Indeed, our analysis shows that SecY, SecD, and YidC are required for the insertion of UbiA TMs other than the cTM (Fig. S3b,d,f). We, therefore, postulate that SecY, and potentially SecD and YidC depletion cause a general biogenesis defect in UbiA that may indirectly prevent the cTM from inserting.”

Lastly, our results that YidC mediates cTM insertion make sense as they fit the sequence constraint that we observe in the C-tails of multispanning membrane proteins. YidC, unlike SecY, cannot translocate long and hydrophilic tail/loop sequences. This point is discussed in our discussion and is shown in references ^{2,20,28,46,58}

Also, as it stands, the studies in Figures 4 and 5 don't really fit with the bacterial studies, because the authors did not show that the studied eukaryotic proteins (ATP6VOC, ENT3 and SGLT1) use an Oxa1 homolog for insertion, although they expect they do.

> Human cells have 3 currently known Oxa1 homologs in the endoplasmic reticulum. We agree that it would be interesting to prove that they insert the cTMs of ATP6VOC, ENT3 and SGLT1, and identify the precise Oxa1 member(s) responsible. However, such studies would be very laborious, and would not add much novel knowledge, especially considering the recent study showing that the Oxa1-related EMC catalyzes a similar activity in human cells (doi: [10.1038/s41594-023-01120-6](https://doi.org/10.1038/s41594-023-01120-6)). With regards to the insertion of wild-type ATP6VOC, ENT3 and SGLT1, this will also be methodologically very difficult, since the C-tails of the wild-type proteins are too short for assaying their insertion by glycosylation (see answer to point 8 below).

We still think that Figs 4 and 5 add much to the paper. The mechanism of insertion, while an important part of this work, is only a part of the discovery of our study. The other major point is the discovery of sequence adaptations of the C-tail that optimize insertion and guide the protein to its correct topology. Showing that mutations that extend the C-tail and sufficiently increase its hydrophilicity cause misinsertion, likely leading to loss of function and genetic diseases in human reveals the clinical implications of our finding. The correlation between C-tail hydrophilicity, misinsertion, and disease, was not known before and is directly related to the title of our paper.

Additionally, there are some technical issues (see below) that need to be addressed to make sure the authors' conclusions are correct. Therefore, while the main message is very interesting and point to an important new role for Oxa1 family members, the conclusions in the submitted paper are still too preliminary to support their conclusions fully.

Other Comments

1.State explicitly how many of the Fig. 1b multispinning membrane proteins in E. coli, yeast and humans have posttranslational C-tails with the carboxy terminus located away from the cytoplasm (Cext) (i.e. must translocate across the membrane).

> We are sorry if this point was not clear before. These number are given as 'n' values in Fig. 1f, and we now clarify that they apply also to Fig. 1e.

2.Please provide the length and the amino acid sequence of the C-tail of UbiA and RcnA.

> We added this in line 147: "UbiA and RcnA, having C-tail sequences 'WHF' and 'R', respectively."

3. For the delta G values to determine the hydrophilicity of the posttranslational C tails in Fig. 1e, Fig. 2e and Fig. 4b, why was the contribution of the peptide bonds not included since this region is translocated across the membrane?

> There are many ways and nominal scales to calculate hydrophilicity. We did not explicitly account for the peptide bond for the following reasons:

(i) The biological hydrophobicity scale (Hessa et al DOI: [10.1038/nature06387](https://doi.org/10.1038/nature06387)), which we use, was generated by studying polypeptides inserted into a biological membrane. Since this scale uses polypeptides, any contribution of the peptide bond is surely included in it.

(ii) The membrane imposes a helical conformation on proteins embedded in it. The biophysical and energetic basis for this is well understood (see <https://doi.org/10.1016/j.jmb.2014.09.014>), and will likely lead to the adoption of a helical conformarion in case a peptide enters the membrane, which is an intermediate state on the path of the C-tail to cross the membrane to the other side. This will lead to the burial of the peptide bond in the helix interior, and thus, the peptide bond is not expected to be directly in contact with the hydrocarbon core.

(iii) Our data in Fig. 2c-e show that sidechain hydrophilicity rather than polypeptide length dominates the sequence requirement for insertion. The observation that the insertion level of three constructs with the same length (4xN, GSGS, HSDS) is different suggests that the peptide bonds are not the dominant factor in determining the sequence requirement.

(iv) The contribution of the peptide bond is very challenging to study experimentally, as it requires chemistry that cannot be encoded genetically. A serious investigation of the role of the peptide bond in this process thus requires a separate study that is beyond the scope of this manuscript.

4. Please provide the equation that was used to calculate the Percent Insertion (translocation) of the C-tails in Fig. 2e and Fig. 3c.

> We apologize for the lack of clarity and thank the reviewer for pointing this out. We now include a full section in our Methods that reads:

“Calculation of the percent of insertion

The percent of insertion is calculated by the percent to which AMS can modify the periplasmic Cys, thereby blocking subsequent PEGylation. In order to calculate it, the percent of PEGylation is quantified by densitometry of SDS-PAGE gels or Western blots, from three samples: an NEM-modified sample (representing the maximal possible blocking of PEGylation, as NEM can modify both cytosolic and periplasmic Cys; the condition controls for inaccessible cysteines which become accessible only after cell lysis and cannot be blocked in whole cells), an AMS-modified sample (representing the blocking emerging from periplasmic Cys only), and an untreated sample (representing the maximal possible PEGylation). Percent insertion is then defined as $\% \text{insertion} = 100 * (1 - \% \text{PEGylation}_{\text{AMS}} / \% \text{PEGylation}_{\text{untreated}}) / (1 - \% \text{PEGylation}_{\text{NEM}} / \% \text{PEGylation}_{\text{untreated}})$.

The controls we include, to quantify the degree of PEGylation in an NEM-treated sample and an untreated sample are critical for the quantification, as some cysteines display less overall PEGylation than others, and some Cys cannot be blocked completely in whole cells before lysis and membrane solubilization, even by NEM which enters the cytosol. We note that the fact that each lane is quantified separately makes it unnecessary to ensure precise equal loading of protein to each lane. However, slight deviations from equal loading may occasionally make it confusing to judge the results upon a quick glimpse, without giving careful thought to the %PEGylation that each lane represents. Notably, the PEGylated band is a bit more smeared than the unPEGylated

protein, since PEG is a polymer with variability in its molecular weight. Thus, PEGylated band that appears relatively faint to the eye typically gives somewhat higher intensity when accurately quantified. We have used this method to accurately quantify TM insertion in previous publications (doi: 10.1038/s41589-019-0356-9; doi: 10.1073/pnas.1706905114).

By eye, the AMS/PEGylation data to assay for translocation of the C-tails (Fig 3b) does not seem to agree with the quantification of insertion results (Fig. 3c). For example, the AMS/PEG studies with the UbiA cTM construct when either SecY or YidC was depleted look quite similar (Fig. 3b). But the quantification of % insertion (Fig. 3c) is quite different.

> Having explained the quantification more carefully, we hope that the reviewer can now agree that upon SecY depletion, the UbiA-cTM appears to be almost 100% inserted: the %PEGylation in the AMS sample is very close to the NEM sample, suggesting that most of the cysteines which are reactive before lysis have been blocked by periplasmic AMS. The behavior under YidC depletion is different - as there is a higher percentage of PEGylation under AMS, suggesting that not all of the C-tails made it to the periplasm for blocking by AMS. In these YidC-relevant samples, there was a slight higher loading of the NEM sample, which might make it a bit confusing for the eye. Most importantly, accurate quantification of many different experiments gave us consistent results regarding the effect of YidC depletion. Gels from five repetitions are given below, along with controls for two constructs under 'Mock' CRISPRi, where YidC is not depleted. In these control samples, UbiA-cTM is inserted, whereas a 4xN extension abolishes insertion

It is not clear how insertion can be accurately determined for the RcnA cTM constructs since so little of the construct is capable of being modified by PEGylation even when no AMS is added (see Fig. 3b).

> The degree of PEGylation is indeed relatively low for RcnA-cTM. The reason for this is that a significant fraction of the GFP-SUMO-cTM protein gets cleaved, yielding a free GFP-SUMO harboring no cTM nor a cysteine. We have now rerun the samples so that the separation of these two protein forms is more evident. We have also changed the legend to Fig. 3b accordingly. It is now more clearly evident that the full GFP-SUMO-cTM construct is quite efficiently PEGylated, allowing us to quantify the level of insertion. Notably, this trend was also visible in the former Fig. 3f, which was now removed and replaced by another result.

5. The authors state that the C-terminal tail translocates slowly for cTM-4NC because they see more AMS modification at 20 min compared to 2 min. This result may not have anything to do with insertion rate but simply be due to the fact AMS modification of the cysteine in the C-tail was not complete at 2 mins, and needed more time. Actually, it is not clear to me how you can look at the rate of insertion by looking at chemical amounts of the protein by western blotting. Typically, one looks at the translocation rate of newly synthesized proteins by doing a pulse chase experiment (radiolabel cells with S35 methionine (for example) for 2 min with no chase or with a 20 min chase with non radioactive methionine).

> We thank the reviewer for making this point, which made us re-evaluate this result and its interpretation. Indeed, we believe the reviewer is right with regard to the time needed for a Cys to complete its reaction with AMS. Since we cannot unequivocally prove what caused the slow labeling - a slow chemical reaction with AMS, or slow insertion, we decided to remove this result from the paper. Nevertheless - we now include a new result to support our original conclusion that UbiA-cTM with a 4xN extension is slow to insert. See our answer to point 6 below.

6. It is not clear what the results of the co-purification of the UbiA and RcnA His tagged sfGFP-Sumo cTM variants with YidC, SecD, or SecY (or lack of) mean since the authors did not measure the interaction during insertion. Typically, you would want to probe the interactions of the substrate with the translocase/insertase as insertion is occurring, not after the membrane insertion process nor with chemical amounts of the proteins extracted from the membrane.

> We thank the reviewer for this concern and realize that we did not explain this experiment well enough in the text. We have therefore revised the text to explain that we are expressing the protein only briefly (by inducing it for 15 minutes), in order to capture 'young' proteins that may still interact with the insertion machinery. Notably, since we have ongoing protein synthesis, some proteins in our cell will have been just synthesized, while others might be many minutes old and therefore might have left the insertion machinery. Nevertheless, the fact that we can still capture the interactions with the insertion machinery and specifically YidC, and the fact that the extension of the cTMs by 4 asparagines modulates this interaction, suggests that the pulldown experiments are informative in explaining the mechanistic effect of the 4xN extensions. These detected interactions are, therefore, likely reflective of the in vivo interactions.

To examine the time-dependent interactions in greater detail, we now include another experiment (Fig. 3F), where the interactions with YidC are probed by pull-down, either

just upon 15-minute expression or after a 30-minute chloramphenicol treatment. Chloramphenicol stops protein synthesis, allowing us to probe the interaction with YidC for 'older' proteins that should have left the insertion machinery if insertion took place. Indeed, our results show that the UbiA-cTM and RcnA-cTM lose their interaction with YidC when they age. By contrast, extending the UbiA-cTM C-tail by 4 asparagines prolongs its interaction with YidC, supporting the idea that the cTM now cannot be efficiently inserted. The section describing these results has been revised (lines 235-255), and we now include the new Fig. 3f and Fig. S7.

Notably, these results do not stand alone, but they only complement the functional results, showing that a four-asparagine extension abolishes the insertion of cTMs, either in the chimeric GFP-SUMO-cTM constructs, or in the full length proteins (Fig. 2 and Fig. 3b,c). We believe that these experiments give a better molecular understanding of the events that may prevent YidC from translocating a hydrophilic loop. Notably, while this added molecular understanding is not a central aim of our current work, we believe that sharing this finding with the scientific community is valuable, and could promote further research in this area regarding Oxa1-family members and potentially other insertion machineries.

7. The authors might consider studying the insertion of newly synthesized UbiA and RcnA using a short radioactive pulse of S35-methionine and then immunoprecipitate the protein with antibody against the HA tag. The YidC, SecD and SecY depletion studies can be carried out with newly synthesized proteins. The AMS/NEM/PEG method could be used with newly synthesized [35S]-labeled construct to show the translocation of the C-terminal tag is inhibited.

> Radioactive experiments have been extremely useful in the field of protein biogenesis, but they have some limitations. From a scientific standpoint the limitation is minor, they require a minimal growth medium in which E coli cells grow slower, which may have physiological consequences for the cell. From a practical standpoint - these experiments are much more laborious as they require pulldown, and also require a separate set of laboratory equipment, which means that we can process much fewer samples in a single experiment. They are also notably more expensive as they require high amounts of antibodies and radioactive material. They additionally pose some safety risks for our students and environment. We, therefore, try to conduct radioactive experiments only when we have no other alternative.

In this case, we didn't think there was a reason for radioactive experiments: (i) For the insertion/topology studies, our results show that the mutant proteins, in which we increased the hydrophilicity of the C-tail, do not get inserted even after 30 minutes of expression. Surely, these mutants would not get inserted also when they have just been expressed upon a radioactive pulse. (ii) for the pulldown studies - radioactive experiments would offer no benefit because we will not have a way to pulldown only the radioactively-labeled protein, and we will therefore pull the newly synthesized radioactive proteins together with older proteins that were synthesized before adding the radioactive methionine pulse. Nevertheless, our approach of using chloramphenicol,

in the new Fig. 3F (see our answer to point 6), offers a solution to answer the point raised by the reviewer.

I would also recommend as a control that the authors show that the co-translational inserted loop preceding the C-terminal tail is blocked by SecY depletion. To do this, they can make another construct and add a cysteine to this loop, and examine its translocation using the AMS accessibility assay.

> We have now done this experiment and include the results in Fig S3b. Indeed, SecY is required for the translocation of a periplasmic loop that precedes the C-tail. Another control that our SecY depletion indeed functionally depletes SecY was provided in the original manuscript (now Fig. S5).

8. It is necessary in the studies described in Fig. 4 to show the C-tail of the parent wild-type ATP6VOC, ENT3, and SGLT1 is translocated. The authors should show the loop preceding the C-tail of the ATP6VOC, ENT3, and SGLT1 mutants is translocated using the PNGase. A glycosylation site can be introduced in this loop.

> We thank the reviewer for their useful suggestion. A glycosylation analysis of the insertion of the loops preceding the C-tails of ATP6VOC and ENT3 is now included in Fig. S9. As the C-tail of the nonstop mutant of SGLT1 is translocated efficiently, we did not continue to analyze this protein. The loops preceding the C-tails of the wild type proteins and the C-tail mutants of ATP6VOC and ENT3 are similarly glycosylated, suggesting that the insertion defect is specific to the mutant C-tails.

We agree with the reviewer that it would be useful to directly show that the C-tail of the wild type ATP6VOC, ENT3 and SGLT1 is translocated. Unfortunately, for efficient glycosylation, the acceptor site needs to be 12-14 amino acids downstream of the transmembrane domain and at least 6 amino acids upstream of the C-terminus (Nilsson & Von Heijne (1993) J. Biol. Chem. 268, 5798-5801; Bano-Polo et al. (2011) Protein Sci. 20, 179-186). As the C-tails of the wild-type proteins are only 2-4 amino acids long, insertion of an efficient glycosylation tag will essentially produce a C-tail mutant and might change the insertion efficiency of the C-tail. Hence, we had to use the indirect localization analysis (Fig. 5) as a proxy for correct folding of the wild type proteins.

9. Why are there two bands for RcnA in Fig. 2d?

> We do not currently know the answer to this, but it occasionally happens with many proteins. It is possible that RcnA exists in two forms in cells, which have different post-translational modifications, or cleavage. We are positive that both bands represent RcnA because: (i) these bands are not observed in Western blots using an anti-HA antibody when RcnA is not expressed. (ii) AMS affects the PEGylation of both bands in a manner that is consistent across the mutants, indicating that both bands harbor a single Cys and that both also harbor the encoded C-terminal mutations. We have now added a sentence to the legend of Fig. 2d: "Note that RcnA displays two bands of slightly different gel migration. "

Reviewer #3 (Remarks to the Author)

The manuscript by Kalinin et al identifies a key problem in polytopic membrane protein biogenesis, that proteins with short soluble C-termini must integrate their last transmembrane domain post-translationally. This is distinct from each of the preceding TMs. They take a broad approach to studying this problem, using bioinformatics and experimental studies that the hydrophobicity of these proteins has been finely tuned to optimize insertion. Furthermore, they demonstrate that in bacteria the insertion of these proteins is dependent on the insertase YidC. This is an elegant study that addresses the evolution of membrane proteins and provides a new model for how these proteins are synthesized. They demonstrate that this knowledge has implications for disease and protein expression. I have no major concerns about this work.

> We thank the reviewer for their assessment and kind words. We have addressed all of the minor concerns, which greatly helped to improve our manuscript.

Minor concerns:

Introduction: Mim1 should be capitalized.

> Done, thank you for pointing this out.

Results:

Pg. 4 – It would be useful to have more of an explanation for the choice of UbiA and RcnA. Are they representative?

> RcnA and UbiA were chosen based on multiple practical parameters. Briefly, we wanted proteins whose topology predictions are highly confident and can be confirmed by an alphafold model or experimental structure, which do not have many cysteines, which have been studied functionally and therefore have an easy functional assay based on growth, and so on, and belong to different structural folds. In fact, we initially had a third protein, but that protein showed diminished expression when its C-tail was extended, preventing us from studying the insertion of the C-tail of the mutant proteins. We are not sure if they are 'representative' as we cannot be sure what parameters can set the proteins apart with regard to the insertion of the cTM. Nevertheless, with regards to C-tail hydrophilicity, there is little variability in the sequences in the *E. coli* membrane proteome (Fig. 1e), and therefore any selected protein would be representative. We have now included a sentence stating that "We chose two *E. coli* proteins with confident topology predictions to study experimentally". The consideration that "The two proteins are structurally unrelated" (line 148) was included in the original paper.

Pg. 6 – For the results in Fig 4, using terms like most for 5 of 7, while correct, to me is misleading. You should just use 5 of 7, be explicit when the numbers are small.

> We agree. We have now changed 'most' to 'five out of seven' (line 348).

Discussion:

These results will have implications for the expression of membrane proteins. It would be nice to include some thoughts towards that.

> We have inserted the following at the end of our discussion: “These proteins may be prone to inefficient biogenesis even under normal conditions and undergo extensive degradation, as has been observed for a number of MPs³. Our findings also serve to caution against adding tags to these proteins in their C-termini, which may perturb their insertion.”

Figures:

1e&f. have similar scales (0-50). Just make sure that is correct. They are measuring different things.

> We thank the reviewer for their concern. We have double-checked this, and the scales are correct.

2a. For clarity, label which (top/correct insertion or bottom/no insertion) should be PEGylated.

> We could not find a way to explain this in the figure without the explanation being potentially confusing. We instead modified the legend to Fig 2a.

2c. For the GSGSC extension, why is there PEG in the NEM control.

> The NEM-modified sample is an important control, representing the maximal possible blocking of PEGylation, as NEM can modify both cytosolic and periplasmic Cys; the condition controls for inaccessible cysteines which become accessible only after cell lysis and cannot be blocked in whole cells. In this case, there existed a small population of proteins where the Cys was inaccessible, and therefore not blocked by NEM, until lysis and addition of detergent, which made it accessible for PEGylation. See also our response to Reviewer 1 comment 4 above. We now include a full section in our Methods that reads:

“Calculation of the percent of insertion

The percent of insertion is calculated by the percent to which AMS can modify the periplasmic Cys, thereby blocking subsequent PEGylation. In order to calculate it, the percent of PEGylation is quantified by densitometry of SDS-PAGE gels or Western blots, from three samples: an NEM-modified sample (representing the maximal possible blocking of PEGylation, as NEM can modify both cytosolic and periplasmic Cys; the condition controls for inaccessible cysteines which become accessible only after cell lysis and cannot be blocked in whole cells), an AMS-modified sample (representing the blocking emerging from periplasmic Cys only), and an untreated sample (representing the maximal possible PEGylation). Percent insertion is then defined as $\% \text{insertion} = 100 * (1 - \% \text{PEGylation}_{\text{AMS}} / \% \text{PEGylation}_{\text{untreated}}) / (1 - \% \text{PEGylation}_{\text{NEM}} / \% \text{PEGylation}_{\text{untreated}}).$ ”

4a. In this case, I find the violin plots misleading. For such small numbers, the scatter clustering is sufficient to convey the results.

> We agree. We have now eliminated the violins.

4c. What are the multiple bands for SGLT1?

> We do not currently know the answer to this, but it occasionally happens with many proteins. It is possible that SGLT1 exists as two forms in cells, which have different

post-translational modifications, or cleavage. We are positive that both bands represent SGLT1 because: (i) these bands are not observed in Western blots using an anti-HA antibody when wild-type is not expressed, and even when the C-terminal disease-causing mutant of SGLT1 is expressed. (ii) Two bands for SGLT1 have been observed before. For example, see Niu et al. (2022) Nat. Comm. 13: 6440, Fig. 1A; Saponaro et al. (2020) Diabetes 69:902-914, Fig. 3b

Reviewer #2 attachment:

The manuscript by Kalinin et al identifies a key problem in polytopic membrane protein biogenesis, that proteins with short soluble C-termini must integrate their last transmembrane domain post-translationally. This is distinct from each of the preceding TMs. They take a broad approach to studying this problem, using bioinformatics and experimental studies that the hydrophobicity of these proteins has been finely tuned to optimize insertion. Furthermore, they demonstrate that in bacteria the insertion of these proteins is dependent on the insertase YidC. This is an elegant study that addresses the evolution of membrane proteins and provides a new model for how these proteins are synthesized. They demonstrate that this knowledge has implications for disease and protein expression. I have no major concerns about this work.

Minor concerns:

Introduction: Mim1 should be capitalized.

Results:

Pg. 4 – It would be useful to have more of an explanation for the choice of UbiA and RcnA. Are they representative?

Pg. 6 – For the results in Fig 4, using terms like most for 5 of 7, while correct, to me is misleading. You should just use 5 of 7, be explicit when the numbers are small.

Discussion:

These results will have implications for the expression of membrane proteins. It would be nice to include some thoughts towards that.

Figures:

1e&f. have similar scales (0-50). Just make sure that is correct. They are measuring different things.

2a. For clarity, label which (top/correct insertion or bottom/no insertion) should be PEGylated.

2c. For the GSGSC extension, why is there PEG in the NEM control.

4a. In this case, I find the violin plots misleading. For such small numbers, the scatter clustering is sufficient to convey the results.

4c. What are the multiple bands for SGLT1?